# Nanoscale three-dimensional fabrication based on mechanically guided assembly

Junseong Ahn [1,2], Ji-Hwan Ha[1,2], Yongrok Jeong[1,2], Young Jung[1], Jungrak Choi[1], Jimin Gu[1], Soon Hyoung Hwang[2], Mingu Kang [1], Jiwoo Ko[1,2], Seokjoo Cho [1], Hyeonseok Han[1], Kyungnam Kang[1], Jaeho Park[1], Sohee Jeon[2], Jun-Ho Jeong [2] ✉ & Inkyu Park [1] ✉

The growing demand for complex three-dimensional (3D) micro-/nanostructures has inspired the development of the corresponding manufacturing techniques. Among these techniques, 3D fabrication based on mechanically guided assembly offers the advantages of broad material compatibility, high designability, and structural reversibility under strain but is not applicable for nanoscale device printing because of the bottleneck at nanofabrication and design technique. Herein, a configuration-designable nanoscale 3D fabrication is suggested through a robust nanotransfer methodology and design of substrate's mechanical characteristics. Covalent bonding–based two-dimensional nanotransfer allowing for nanostructure printing on elastomer substrates is used to address fabrication problems, while the feasibility of configuration design through the modulation of substrate's mechanical characteristics is examined using analytical calculations and numerical simulations, allowing printing of various 3D nanostructures. The printed nanostructures exhibit strain-independent electrical properties and are therefore used to fabricate stretchable $H_2$ and $NO_2$ sensors with high performances stable under external strains of 30%.

Complex three-dimensional (3D) micro- and nanostructures provide key functions in systems with various sizes and functions, which range from the organs of living organisms to microscale devices (e.g., foldable microelectronics[1], deformable batteries[2], and bioelectronics[3,4]) and nanoscale devices (e.g., chemical/physical sensors[5,6], surface-enhanced Raman spectroscopic substrates[7], and holographic displays[8]). Therefore, methods for accurately designing and fabricating complex 3D structures are of importance for both academia and industry, finding applications in the biological, electronic, mechanical, and optical engineering fields[9]. Given the difficulty of fabricating complex 3D structures using conventional two-dimensional (2D) planar processes such as inkjet/screen printing and lithography/ion-milling-based techniques, recently developed and more advanced 3D fabrication methods

have drawn much attention[10]. In particular, 3D fabrication methods based on mechanically guided assembly (i.e., compressive buckling–based printing) allow the fabrication of complex 3D structures in thin and curvilinear forms and offer the advantages of wide material scope (e.g., metals, ceramics, and polymers), high designability, precise controllability, scalability, and structural reversibility under strain, thus holding great promise for next-generation printing[11,12]. In this case, a 2D structure (divided into bound and suspended sites for selective bonding) is initially printed on a prestretched elastomer substrate using conventional 2D fabrication techniques, and the suspended sites are then shape-morphed into a 3D structure under the action of compressive buckling generated upon the release of pre-strain[13–16]. This process requires one to answer two

[1]Department of Mechanical Engineering, Korea Advanced Institute of Science and Technology (KAIST), Daejeon 34141, Republic of Korea. [2]Department of Nano Manufacturing Technology, Korea Institute of Machinery and Materials (KIMM), Daejeon 34103, Republic of Korea. ✉e-mail: jhjeong@kimm.re.kr; inkyu@kaist.ac.kr

questions related to the definition of 3D fabrication and the realization of complex 3D structures, namely How does one design and predict the final 3D structure? and How does one manufacture the designed 3D structure? The successful production of 3D structures and their stable printing requires strong adhesion between the bound sites and the substrate to provide rigid support during the generation of buckling. However, current techniques allowing 2D printing on stretchable elastomeric substrates suitable for 3D fabrication have size limitations, with stable adhesion guaranteed only for sizes above tens of micrometers because of low adhesion force at the nanoscale[17]. In addition, the current configuration design method is based on the patterning of an adhesive layer that will become bound sites on the printing materials before transfer. However, applying this conventional method to extremely thin adhesives (i.e., self-assembled monolayers) and nanopatterned printing materials used in 2D nanotransfer printing is challenging. Therefore, the design and control of nanoscale buckling configuration (e.g., beam shape, direction, deflection, and mode) are much more difficult. Thus, the realization of configuration-designable nanoscale 3D fabrication based on mechanically guided assembly remains challenging, and the demand for 3D nanodevices for applications such as gas sensors, electrodes, thermoacoustic speakers, and optical devices remains unmet[18] (Supplementary Notes 1).

Herein, we develop a mechanically guided assembly–based nanoscale 3D fabrication method allowing the rational design of buckling configurations and the printing of both bound and suspended sites with a width as small as 50 nm. To achieve this goal, we focus on design and manufacturing as the two main parts of 3D fabrication. Regarding the manufacturing part, 2D nanotransfer printing on an elastomer substrate is realized using a covalent bonding–assisted adhesive with a sub-10-nm thickness and the selective etching of a nanopatterned mold to reduce the adhesion between the target material and the mold. Regarding the design part, micropatterning is used to modulate the elastomer's mechanical characteristics and thus enable the rational design and prediction of the printed buckling configuration (i.e., direction, deflection, and mode) for a given 2D pattern. The developed approach is used to fabricate 3D nanostructures with diverse configurations and compositions, and the electromechanical properties of these structures are characterized in detail. This is the report of a configuration-designable nanoscale 3D fabrication process featuring nanoscale bound and suspended sites (Supplementary Fig. 1 and Supplementary Notes 1). Finally, the developed technique is used to print a stretchable Pd-based $H_2$ sensor and an $In_2O_3$-based $NO_2$ sensor, both of which show higher sensitivity than their conventional counterparts and maintain stable performances under external strain because of their buckled 3D nanostructures.

## Results

### Overall 3D fabrication concept

According to the overall concept of configuration-designable nanoscale 3D fabrication (Fig. 1), the main parameters determining the configuration of compressive strain–induced beam buckling are the shape of the 2D pattern and buckling direction, deflection, and mode. The 2D pattern shape is set during the 2D printing step, whereas the other parameters are related to the input conditions of buckling. Specifically, buckling direction is determined by the direction of the initial deformation (which, in turn, is affected by the beam

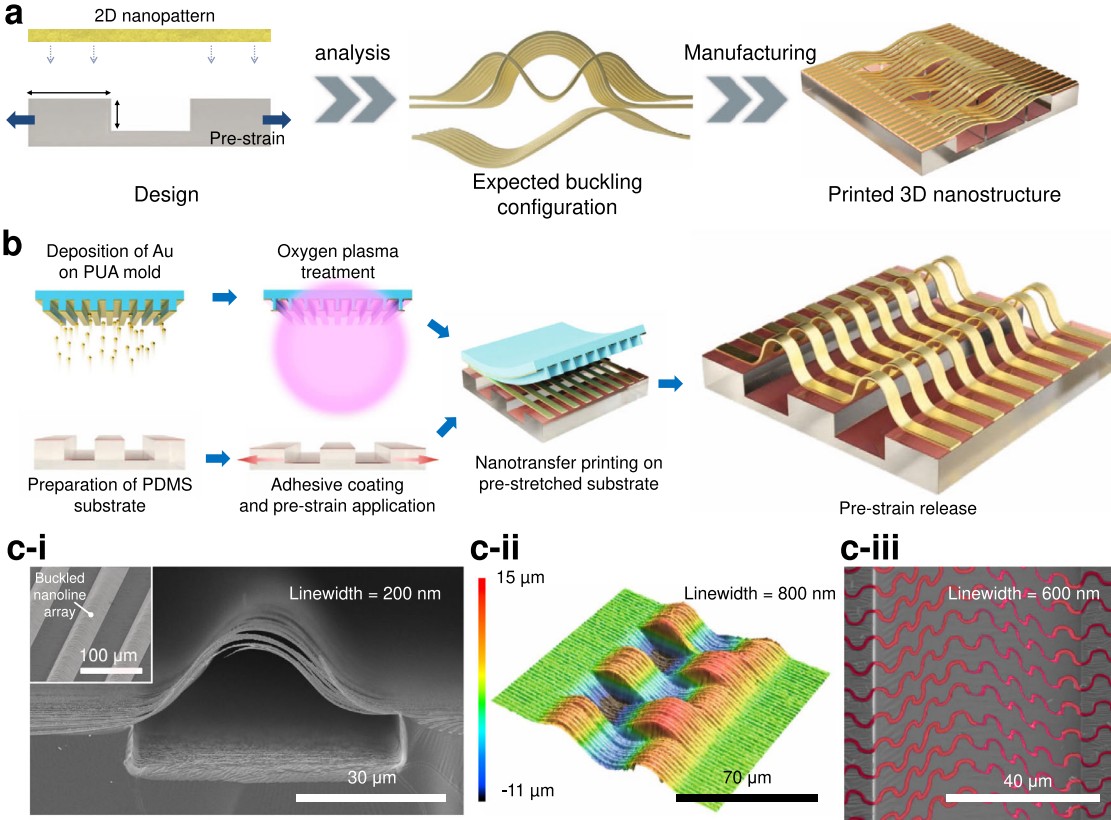

**Fig. 1 | Concept and realization of configuration-designable nanoscale 3D fabrication. a** The overall scheme of 3D fabrication including design and manufacturing steps. **b** Detailed fabrication process of 3D fabrication: Target material deposition, treatment with $O_2$ plasma, preparation of micropatterned elastomer mold, spin-coating of the thin adhesive layer, application of pre-strain to the substrate, nanotransfer printing, and pre-strain release. **c-i** Side-view scanning electron microscopy image of a buckled nanoline array with a linewidth of 200 nm and a thickness of 80 nm; the inset shows the corresponding large-area bird's-eye view image. **c-ii** Confocal laser scanning microscopy image of nanoline arrays with buckling modes 2 and 3 featuring a linewidth of 800 nm and a thickness of 100 nm. **c-iii** Colored scanning electron microscopy image of a buckled serpentine array with a linewidth of 600 nm and a thickness of 100 nm.

imperfections or mechanical guidelines), buckling deflection is directly related to the magnitude of compressive strain, and buckling mode is related to the boundary constraints. Herein, we suggest that the abovementioned parameters can be controlled by designing the mechanical characteristics of the micropatterned substrate, e.g., surface strain and Poisson's effect. In the proposed method (Fig. 1a), the 3D buckling configuration of 2D nanostructures can be predicted before printing and fabricated into the desired shape, which is an essential characteristic of nanoscale 3D fabrication. A detailed explanation of the above principle is provided in the following sections. Figure 1b shows the details of the nanoscale 3D fabrication process that is divided into three steps. In the first step, the target materials are deposited on the nanopatterned poly(urethane acrylate) (PUA) mold using an e-beam evaporator, and oxygen plasma treatment is then applied to facilitate nanotransfer printing. In the second step, an adhesion promoter, namely N-[3-(trimethoxysilyl)propyl]ethylenediamine, is coated on the micropatterned polydimethylsiloxane (PDMS), and pre-strain is applied. In the third step, the target materials with the

nanopattern are transferred to the PDMS substrate and printed as 3D nanostructures by releasing substrate pre-strain. The developed process can be used to obtain diverse 3D nanostructures, e.g., Fig. 1c-i shows a side-view scanning electron microscopy (SEM) image of a buckled nanoline array with a linewidth of 200 nm and a thickness of 80 nm, while the inset shows the corresponding large-area bird's-eye view image. Figure 1c-ii presents a confocal laser scanning microscopy (CLSM) image of a nanoline array with buckling modes 2 and 3, while Fig. 1c-iii shows the corresponding image of a buckled nano-serpentine array. Detailed images of the substrate and mold as well as enlarged views of buckled structures are presented in Supplementary Fig. 2.

**Versatile nanotransfer printing on the elastomer substrate**
Figure 2 shows the mechanism of 2D nanotransfer printing on the elastomer substrate and the related fabrication details. Notably, the nanotransfer method used for nanoscale 3D fabrication requires (1) the adhesion force between the bound site and the elastomer to be

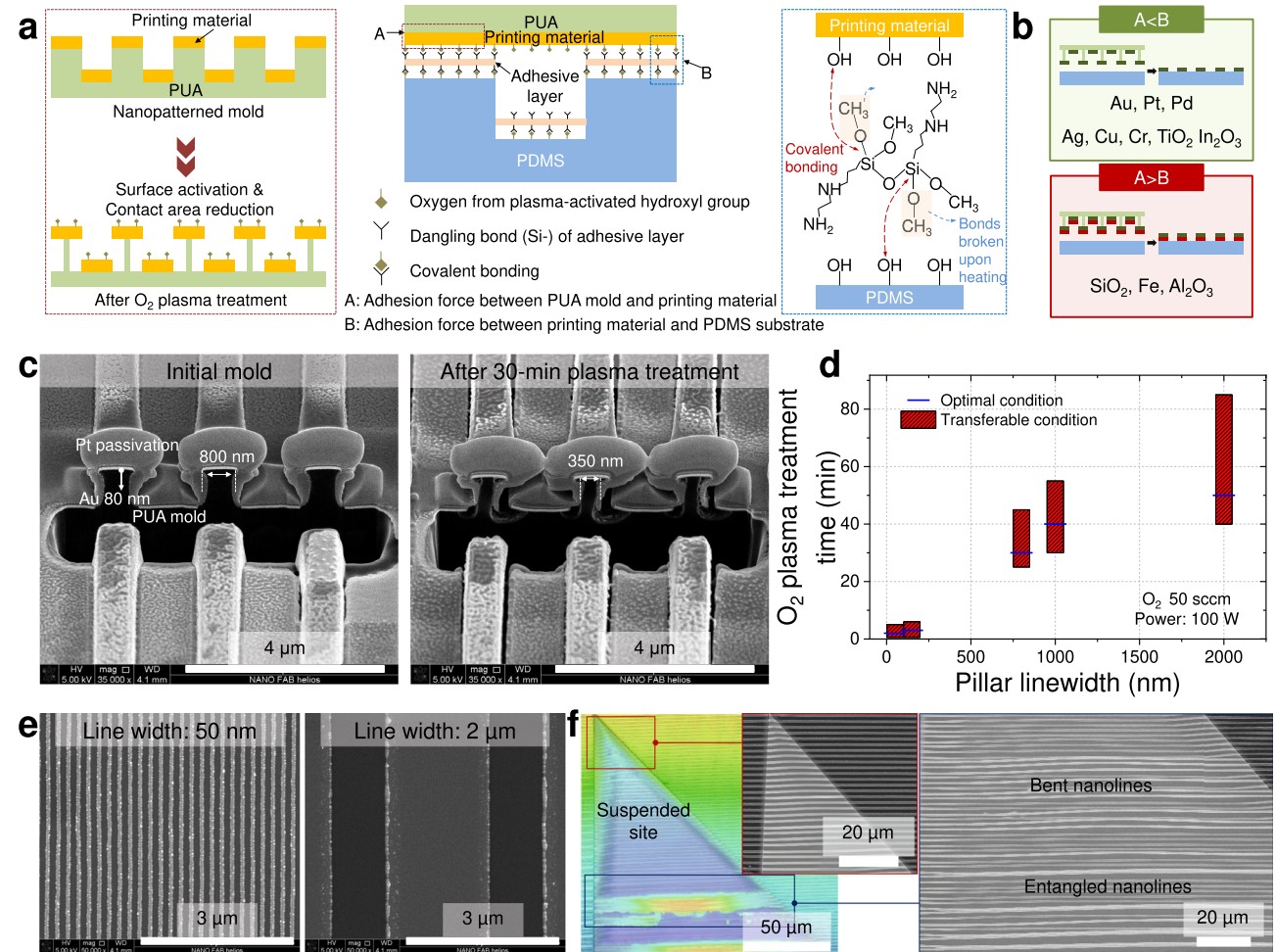

**Fig. 2 | Mechanism and fabrication details of two-dimensional nanotransfer printing on the elastomer substrate. a** Schematic mechanism of nanotransfer. The area enclosed by a red dotted line shows the effect of $O_2$ plasma treatment on the mold with the target material, and the area enclosed by a blue dotted line shows the chemical reactions of the adhesion promoter during nanotransfer printing. **b** Schematic effects of material properties on material transferability during nanotransfer printing. Materials with weak intrinsic adhesion to the mold can be transferred directly (area enclosed by the green line), while those with strong intrinsic adhesion require a supporting layer (e.g., Au) for transfer (area enclosed by the red line). **c** Side-view scanning electron microscopy images of the mold with the target material before and after 30-min treatment with $O_2$ plasma. **d** Effects of the

linewidth of the mold nanopattern on the possible range and optimum of $O_2$ plasma treatment time suitable for transfer. Here, a transferable condition is defined as a condition in which 50% or more of the target region is transferred and an optimal condition is defined as a condition in which 80% or more of the target region is transferred. **e** Scanning electron microscopy images of nanoline patterns with linewidths of 50 nm and 2 μm nanotransfer-printed on a flat elastomer substrate. **f** Confocal laser scanning microscopy and scanning electron microscopy images of a nanoline array with a linewidth of 800 nm nanotransfer-printed on the micropatterned elastomer substrate. The triangular region of the confocal laser scanning microscopy image denotes the suspended (trench) part.

sufficient for supporting the suspended site, (2) the adhesive layer to be sufficiently thin not to affect the printing process, and (3) the selective bonding of the bound site only. Figure 2a presents the three strategies employed to meet these requirements. An ultra-thin (sub-10-nm) layer of an organic ligand was used to promote adhesion and enable strong covalent bonding between the target material and the elastomer substrate[19]. Moreover, $O_2$ plasma treatment was used to selectively etch the nanopatterned PUA mold and activate the hydroxyl groups of the target material. Etching reduced the area of the mold contacting the target material and thus weakened the adhesion between them, while hydroxyl group activation enabled covalent bonding between the target material and the adhesive layer on the substrate. Finally, microtrenches were formed on the substrate surface to facilitate selective bonding, which is additionally related to buckling configuration control. The effects of each strategy were verified experimentally (Supplementary Fig. 3) to reveal that the thin organic ligand promoted the adhesion of the target material to the elastomer substrate, while $O_2$ plasma treatment mainly promoted the delamination of the target material from the PUA mold. The adhesion force during final transfer was sufficient to support the suspended site by the bound site (Supplementary Fig. 3). Depending on their intrinsic properties, most materials (e.g., Au, Pt, Pd, Ag, Cu, Cr, $TiO_2$, and $In_2O_3$) could be directly transferred to the elastomer substrate, whereas the transfer of certain materials strongly adhering to the PUA mold (e.g., $SiO_2$, $Al_2O_3$, and Fe) required an additional supporting layer (e.g., Au) (Fig. 2b and Supplementary Table 1). Notably, our method featured a broad scope of transferable materials (e.g., metals and ceramics) and was applicable to the transfer of multilayer structures (Supplementary Fig. 4). Therefore, materials easily oxidized during treatment with $O_2$ plasma (e.g., Ag and Fe) could be transferred without oxidation or corrosion via encapsulation with noble metals before printing (Supplementary Fig. 5). It is also expected that these materials can be transferred with $H_2$ plasma treatment without oxidation[20].

Subsequently, we evaluated fabrication details and conditions, revealing that with increasing time of PUA mold treatment by $O_2$ plasma, the PUA pillars supporting the target material become thinner, and the target material eventually collapses (Fig. 2c and Supplementary Fig. 6a). Therefore, optimal conditions depending on the pattern size have to be established, as excessive etching destroys the pattern arrangement and causes the transfer of the unwanted layer deposited on the trench (Supplementary Fig. 6b). The results of transfer condition optimization show that the optimal etching time corresponds to a three-fold reduction in the supporting pillar width and is highly dependent on the target material, i.e., on its intrinsic adhesion properties (Fig. 2d and Supplementary Table 1). The optimal conditions allowed the transfer of diverse nanopatterns with different widths and thicknesses to the elastomer substrate (Fig. 2e and Supplementary Figs. 7 and 8) and were also valid for transfer to a micropatterned elastomer substrate (Fig. 2f). When a nanoline array with a linewidth of 800 nm and a thickness of 100 nm was transferred, the original structure was well maintained up to a suspending length of 100 μm, whereas bent or entangled nanolines were observed above this limit.

**Design and prediction of the nanoscale buckling configuration**
Subsequently, a method for controlling the buckling configuration (i.e., direction, deflection, and mode) through the design of the substrate's mechanical characteristics was developed (Fig. 3a, b) and characterized using the parameters including $w_{pillar}$ (pillar width), $t_{pillar}$ (pillar thickness), $t_{substrate}$ (substrate thickness), $\varepsilon_{pillar}$ (average surface strain of the pillar), $\varepsilon_{trench}$ (average surface strain of the trench), and $\varepsilon_{substrate}$ (average strain of the substrate, which equals the external strain or pre-strain applied to the substrate) (Supplementary Figs. 9 and 10 and Supplementary Notes 2). As shown in Fig. 3c, the buckling direction is determined by the direction of the initial deformation and

is affected by the pillar edge shape. Notably, a micropatterned PDMS substrate with smoothed pillar edges can be fabricated when the precursor is poured into a micropatterned SU-8 mold without high-vacuum conditions. When the substrate is subjected to an external strain, the pillar edges rise because of the pillar bending caused by shear stress applied on the bottom of the pillar (Fig. 3c-ii and Supplementary Fig. 11). Therefore, when 3D fabrication is conducted at low pre-strain ($\varepsilon_{pillar} < 2.5\%$), the beam is guided along the smoothed edge, and buckling occurs in the downward direction. On the contrary, when 3D fabrication is performed at high pre-strain ($\varepsilon_{pillar} > 2.5\%$), buckling occurs in the upward direction (Fig. 3c-iii and Supplementary Fig. 12). Figure 3d shows the relationship between the deflection extent and printing conditions in terms of $\varepsilon_{trench}$. As reported previously, the deflection of the buckled beam fabricated using the compressive strain generated during the release process of the pre-stretched substrate is only determined by the pre-strain applied to the substrate (a schematic illustration of the buckled beam is given in Supplementary Fig. 13)[21]:

$$\text{Governing equation}: E_{beam}I_{beam}\frac{d\theta}{ds} = P(w_B - w), \qquad (1)$$

$$\text{Solution}: w_{beam}^{max} = 2z_{beam}^B = \frac{\beta L_{beam}}{K(\beta)}, \qquad (2)$$

$$\varepsilon_{pre} = \frac{1}{2\frac{E(\beta)}{K(\beta)} - 1} - 1. \qquad (3)$$

In these equations, $L_{beam}$ is the length of the beam, $E_{beam}$ is the Young's modulus of the beam, $I_{beam}$ is the moment of inertia of the beam, $\theta$ is the angle which the tangent at a point of the curved beam AB makes with the x-axis, s is the distance along the axis of the curved beam from A, P is the force applied at B by the adjacent part of AB, w is deflection in the z-direction, $w_B$ is deflection at B, $\beta$ is defined as $\sin\frac{\alpha}{2}$ ($\alpha$ is $\theta$ at B), $\varepsilon_{pre}$ is the amount of pre-strain applied to the substrate, $K(\beta)$ is the complete elliptic integral of the first kind, and $E(\beta)$ is the complete elliptic integral of the second kind.

As shown in Fig. 3d-ii and Supplementary Fig. 14, the experimental results regarding the maximum beam deflection as a function of $\varepsilon_{trench}$ (in this research, the trench strain is the same as the pre-strain ($\varepsilon_{pre}$) in Eq. (3)) were well-matched with the results of finite element method (FEM) simulation and analytical calculations. In addition, as shown in Fig. 3d-iii, the configuration of the single beam (measured using CLSM (Supplementary Fig. 15)) was well-matched with the results of the FEM simulation. Here, for the above analysis, the top surface of the pillar was assumed as flat under the applied strain. Because the amount of edge deformation under strain is relatively minor compared to the pillar width (originated from the relatively small $\varepsilon_{pillar}$) (Supplementary Fig. 16); it affected the buckling direction only, not the overall deflection configuration. Subsequently, we investigated the buckling mode (Fig. 3e), which describes the shape of the beam when the buckling occurs (to intuitively explain it, mode 1 has no inflection point, mode 2 has one inflection point, and mode 3 has two inflection points). Generally, buckling with mode 1 occurs most frequently because the required critical force increases as the mode number increases (i.e., when a gradually increasing compressive force is applied to a beam, buckling with mode 1 occurs first before generating mode 2 or 3 under higher compressive force). However, buckling with higher mode can be controlled by adding mechanical constraints hindering buckling with lower mode. Thus, we developed a design and fabrication method for controlling the beam constraints by choosing appropriate printing boundary conditions, enabling the design of buckling mode. For width ($w_{pillar}$) above 10 μm, the bound site enables beam fixation, as

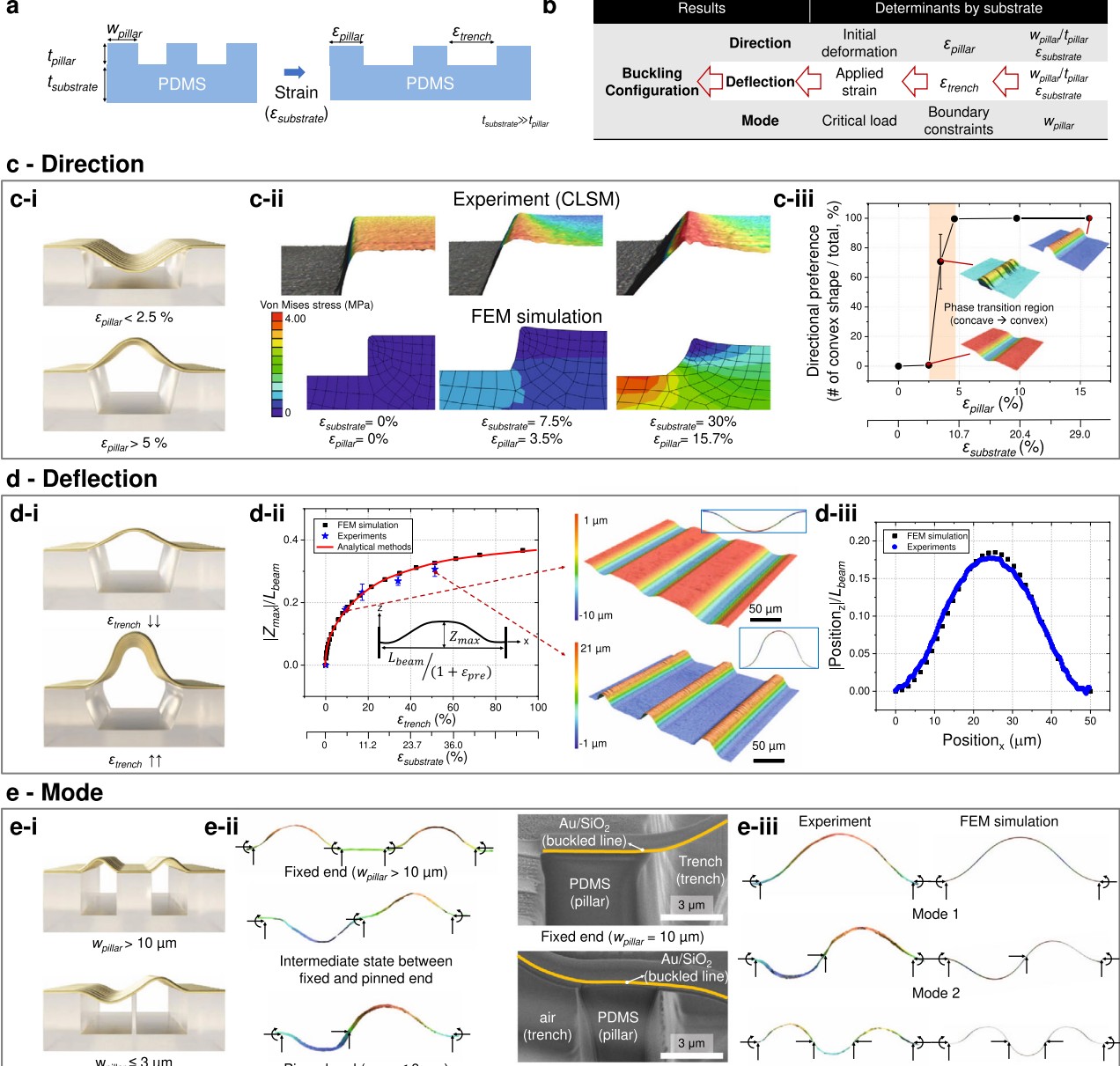

Fig. 3 | Design of buckling configuration and the related fabrication mechanism. a Definition of the design parameters of the substrate. b Determinants by substrate microstructure for controlling the three components of buckling configuration (i.e., direction, deflection, and mode). c Mechanism of buckling direction control. c-i Schematic effect of pillar strain on buckling direction. c-ii Confocal laser scanning microscopy images and finite element method simulation results showing the effects of pillar strain on the pillar edge shape. c-iii Number of convex shapes (upward buckling) depending on pillar strain. d Mechanism of buckling deflection control. d-i Schematic dependence of buckling deflection on trench strain. d-ii Effects of trench strain on the maximal $z$-directional deflection of the buckled beam determined using experiments, theoretical analysis, and finite element method simulation. d-iii Experimental and simulated results of overall beam configurations under the applied printing conditions (applied external strain ($\varepsilon_{substrate}$) = 5%). e Mechanism of buckling mode control. e-i Schematic of the dependence of buckling mode on the boundary constraints. e-ii Confocal laser scanning microscopy and scanning electron microscopy images of the buckled beam acquired for different boundary constraints with various pillar widths. e-iii Experimental and simulation results obtained for buckled beams with the designed modes printed using different pillar sizes and arrangements.

the thick pillar limits both beam displacement and beam rotation. In contrast, $w_{pillar} < 3\,\mu m$ corresponds to a pinned condition that limits only beam displacement and allows beam rotation, as fixation by a thin (compared with beam length) pillar is similar to point fixing (Fig. 3e-ii). With these design parameters, we fabricated beams with various buckling modes (e.g., modes 1, 2, and 3) and observed close agreement with the simulation results (Fig. 3e-iii). In conclusion, we found that the buckling configuration of the nanoscale beam can be designed and controlled through the design of substrate surface microstructure and substrate pre-strain.

## 3D fabrication of diverse 3D nanostructures

The practicality of the nanoscale 3D fabrication method was verified by printing various 3D nanostructures (Fig. 4). Among the designable parameters (i.e., shape of precursor 2D nanopattern, buckling direction, buckling deflection, and buckling mode), the nanopattern shape was examined first. Figure 4a shows the 3D nanostructures printed using different precursor 2D nanopatterns (e.g., vertical lines, diagonal lines, mesh, and serpentine) and a line-patterned elastomer substrate. Regardless of the 2D nanopattern shape, the corresponding 3D nanostructures were successfully printed on the

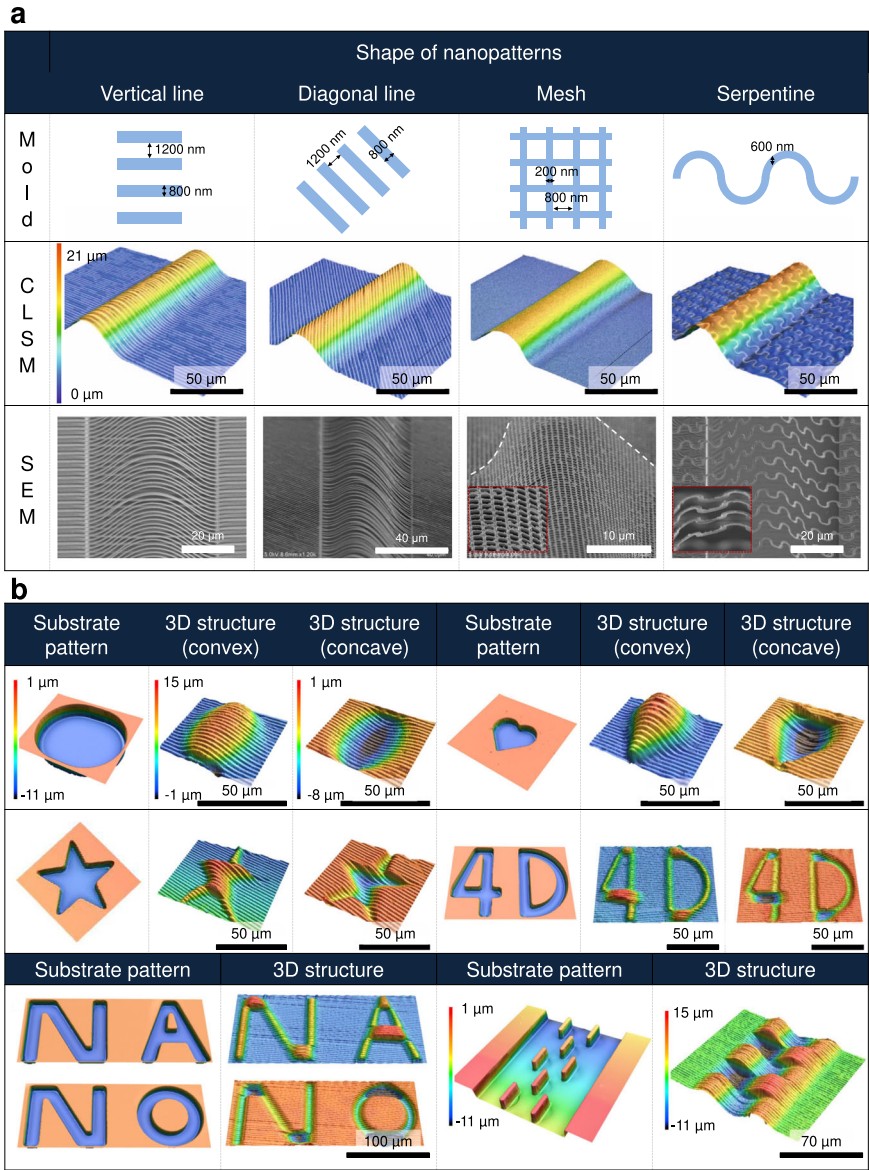

**Fig. 4 | Results of mechanically guided assembly–based 3D fabrication under various conditions. a** Schematic illustration of precursor two-dimensional nano-patterns (e.g., vertical lines, diagonal lines, mesh, and serpentine) and microscopic images of three-dimensional nanostructures printed using the corresponding nanopatterned molds. A white dashed line in nanomesh pattern is used to better visualize the curvature of the nanomesh caused by buckling. **b** Confocal laser scanning microscopy images of elastomer substrates with different micropatterns and corresponding images of three-dimensional nanostructures printed on these substrates. Three-dimensional nanostructures with convex (upward buckling) and concave (downward buckling) shapes were printed by subjecting the substrate to different pre-strains (i.e., $\varepsilon_{substrate} \approx 20\%$ and 5% for convex and concave shapes, respectively), which is related to pillar strain ($\varepsilon_{pillar}$) and the resultant pillar edge shape. The structure displayed in the bottom right is a complex structure with buckling modes 2 and 3 fabricated with a complex substrate of thin and wide bound sites.

elastomer substrate. Then, buckling configuration designability was demonstrated for a specific 2D nanopattern (nanoline pattern with a linewidth of 800 nm), as shown in Fig. 4b. 3D fabrication was applied to substrates with various micropatterns (e.g., ellipse, heart, star, 4D, and NANO), and in each case, 3D nanostructures with convex (upward buckling) and concave (downward buckling) shapes were printed by applying different pre-strains to the substrate (e.g., $\varepsilon_{substrate} \approx 20\%$ and 5% for convex and concave shapes, respectively). In addition, the substrate was also designed to have a complex of thin and wide bound sites (i.e., a complex of pinned and fixed ends) to print a complex structure with buckling modes 2 and 3. The results of these experiments confirmed that the developed strategy allows one to control the configuration of nanoscale buckling and to print various 3D nanostructures.

## Gas sensor application

To demonstrate the applicability of nanoscale 3D fabrication, we used it to fabricate highly sensitive and stretchable gas sensors. Considering the growing demand for wearable electronics, stretchable devices such as pressure sensors[22–24], strain sensors[25,26], and heaters[27] have received increased attention. Among the various strategies for realizing wearable electronics, strain-insensitive device fabrication is particularly important for ensuring the precise functioning of electronic components[28]. However, for gas/chemical sensors, the realization of strain-insensitive properties is difficult[29], as the main constituent materials such as metal oxide semiconductors and noble metals are brittle and cannot be encapsulated by a stretchable matrix because of the need to expose the sensing materials to gaseous species. Moreover, the complex nanostructures with high surface-area-to-volume

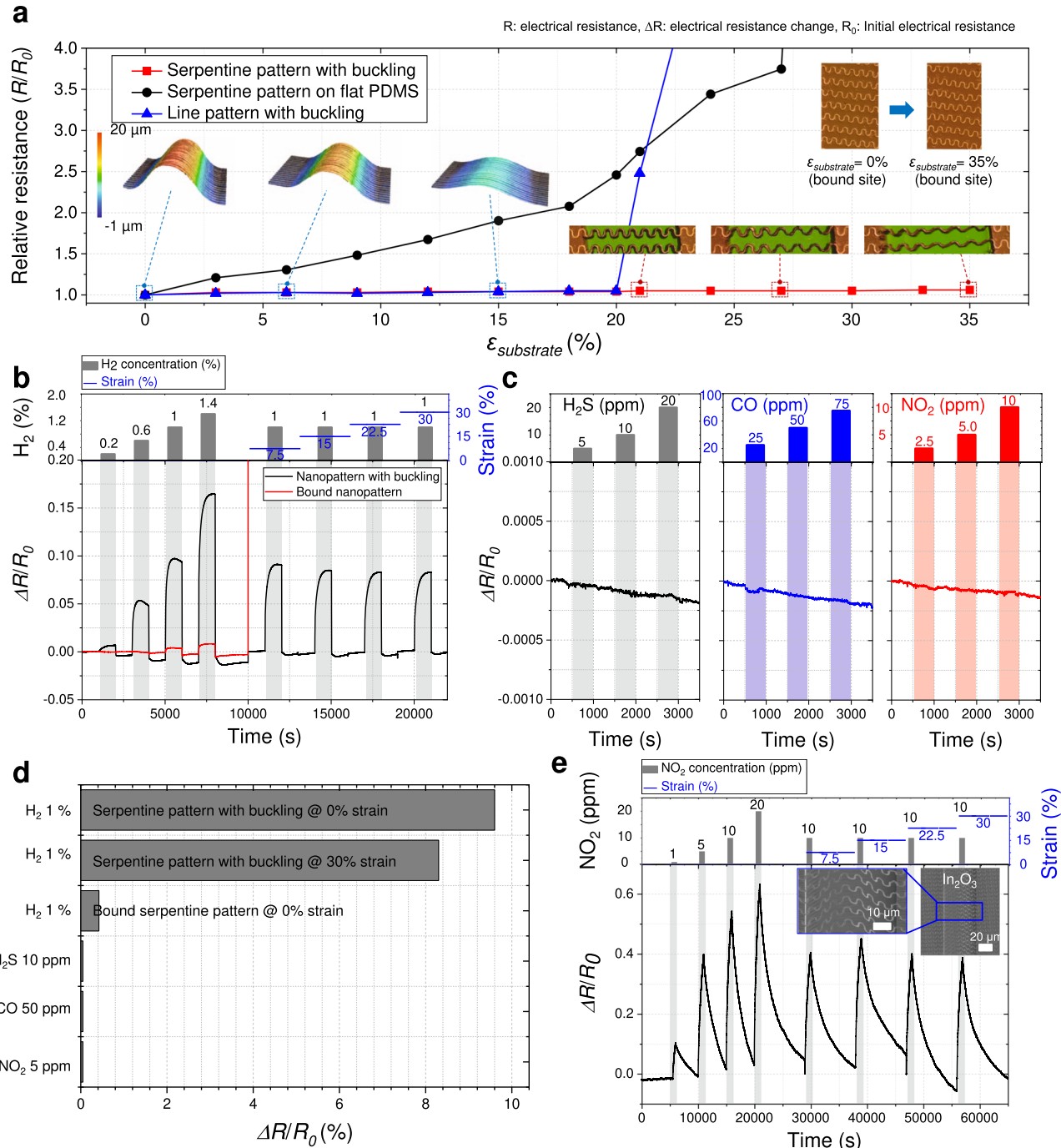

**Fig. 5 | Electromechanical characteristics and performance of gas sensors.**
**a** Relative electrical resistance of Pd nanostructures with/without buckling as a function of external strain. **b** Relative resistance of a Pd nano-serpentine pattern with buckling and that of a bound Pd nanopattern as functions of $H_2$ level and external strain. **c** Relative resistance of the Pd nano-serpentine pattern with buckling as a function of $H_2S$, CO, and $NO_2$ levels. **d** Overall results of gas tests. The selectivity test was performed only for the sensor based on the Pd nano-serpentine pattern with buckling. **e** Relative resistance of $In_2O_3$ nano-serpentine pattern with buckling as a function of $NO_2$ level and external strain. The colored shading regions (gray, blue, and red) mean injection regions of each gas.

ratios required for securing high sensitivity are difficult to fabricate on stretchable substrates.

To address these limitations, we employed nanoscale 3D fabrication to fabricate strain-insensitive gas sensors. Initially, the electromechanical characteristics of the 3D printed nanostructures were evaluated to fabricate rationally designed gas sensors. When a strain was applied to a 3D nanostructure fabricated using the nano-serpentine pattern and a pre-strain of 20%, the initial resistance was maintained (resistance change <5%) until an external strain of 35% (Fig. 5a, Supplementary Fig. 17, and Supplementary Notes 3).

Subsequently, the 3D nanostructure based on the nano-serpentine pattern was used as a Pd-based $H_2$ sensor, offering three advantages over conventional bound nanopatterns in view of its unique buckled structure. Specifically, the utilization of all four sides of the nanoline pattern increased the active surface area, the effect of constraining the volume expansion of Pd by the bound substrate during the reaction was eliminated[30], and strain-insensitive sensing abilities were achieved as mentioned above. Figure 5b shows the gas sensing performances of buckled and bound nanopatterns under identical conditions. When both sensors were exposed to $H_2$, the buckled nanopattern showed

27- and 20-fold higher responses than the bound nanopattern at $H_2$ levels of 1 vol% and 1.4 vol%, respectively. Notably, the buckled nanopattern maintained its response to 1 vol% $H_2$ with a variance of 12.0% when an external strain of up to 30% was sequentially applied, while the bound nanopattern was broken under the action of an external strain of 7.5%. In addition, the experiments for sensing performance after a cyclic test with the repeated loaded/unloaded strain were conducted as shown in Supplementary Fig. 18. The fabricated sensor was not damaged during the cyclic test of the repeated stretching/releasing, and the variation of 11.7% (i.e., the difference between sensor responses to the $H_2$ gas during the stretching/releasing cyclic test) is still within the original sensor variation of 12.0% (i.e., the difference between sensor responses to the $H_2$ gas during the repeated exposure to the air/H2 gas). Figure 5c presents the selectivity of the buckled nanopattern–based sensor. The negligible responses (<0.01%) of the stretchable $H_2$ sensor fabricated by 3D fabrication to 10 ppm $H_2S$, 50 ppm CO, and 5 ppm $NO_2$ were indicative of its superior sensitivity and selectivity, revealing that this sensor could be effectively used even under external strain (Fig. 5d).

In addition, by adopting different printing materials (Fig. 2b and Supplementary Table 1), we fabricated other types of stretchable gas sensors. For example, a buckled $In_2O_3$–based sensor (Fig. 5e) was used to detect $NO_2$, which can damage the human respiratory tract and increase the vulnerability to and the severity of respiratory infections and asthma[31]. Similar to the $H_2$ sensor, it maintained the response to 10 ppm $NO_2$ with a variance of 12.5% when an external strain of up to 30% was sequentially applied. It reveals that various gas sensors with different detection gases can be fabricated using the developed nanoscale 3D fabrication. A detailed explanation of the working mechanisms of each gas sensor is described in Supplementary Notes 4.

## Discussion

A strategy for mechanically guided assembly–based 3D fabrication was developed and shown to enable the printing of nanoscale 3D structures with designable configurations. Nanotransfer to the elastomer substrate was achieved by modulating the adhesion force between the polymer mold, target material, and the elastomer characteristics, and the relationship between the substrate's mechanical properties and the final buckling configuration was studied to achieve a rational design of fabricated 3D nanostructures. Subsequently, diverse 3D nanostructures with different configurations were printed using various 2D nanopattern shapes, buckling directions, buckling deflections, and buckling modes to demonstrate the feasibility of the suggested process and the related design diversity. The printed structures showed strain-insensitive electrical characteristics due to the designed buckling structure and were used to fabricate Pd- or $In_2O_3$-based high-performance stretchable gas sensors.

Despite the above success, some challenges remain to be addressed. First, the longitudinal size of the printed structures needs to be further decreased, as this study has only discussed nanoscale width and thickness, while the length was beyond the microscale. To handle this issue, one needs to develop methods of nanoscale alignment during 3D fabrication, as this alignment is required to precisely design and control the final 3D structure. Second, the commercialization of the nanoscale 3D fabrication process requires the realization of an inverse design process and printing under omnidirectional strain. Herein, we discussed the relationship between the fabrication parameters and the final buckling configuration for printing under unidirectional strain using analytical calculations, FEM simulations, and experiments. We expect that an inverse design tool including omnidirectional strain analysis can be developed in the near future based on these findings and previous studies on microscale printing[32].

Nonetheless, the results presented herein should help resolve the bottleneck related to the printable size of mechanically guided assembly and pave the way to nanoscale 3D fabrication realization and commercialization for the fabrication of optical devices, physical/chemical sensors, catalysts, and bioelectronics.

## Methods

### The procedure of mechanically guided assembly–based 3D fabrication

The developed 3D fabrication process featured the following three steps.

(1) *Nanopatterned mold fabrication*. A Si master was prepared by krypton fluoride lithography to fabricate sacrificial nanopatterned polymer films. For polymer mold replication, RM-311 polyurethane resin (Minuta Technology Co., Ltd., Korea) was poured onto the prepared Si master and covered by a UV-transparent polyethylene terephthalate film. Proper pressure was applied to the resin using a hand roller to facilitate the complete penetration of the resin into the master nanopatterns, and the resin was subsequently cured under UV light of a wavelength of 300–400 nm with a peak wavelength of 365 nm and separated from the master. This process could be conducted repeatedly, and various polymer molds with different nanopatterns could be fabricated using different Si masters[33,34].

(2) *Micropatterned substrate fabrication*. For substrate surface micropatterning, SU-8 was patterned on a Si wafer by photolithography using a stepper (MDA-8000B, Midas System, Korea) and then treated with a self-assembled monolayer (trichloro(1H, 1H, 2H, 2H-perfluorooctyl)silane, Sigma-Aldrich, USA) to facilitate the separation of the elastomer from the SU-8 mold. Subsequently, a commercial elastomer precursor (PDMS, SYLGARD 184, Dow Chemical Co., USA) was poured on the SU-8 mold and cured at 50 °C for 10 h. The micropatterned elastomer was separated from the SU-8 mold and used as a substrate.

(3) *Nanotransfer printing on elastomer substrate with pre-strain and release*. Target materials such as Au, $SiO_2$, Pd, Pt, Ag, Cu, Cr, $TiO_2$, Fe, $Al_2O_3$, and $In_2O_3$ were deposited on the nanopatterned PUA mold using an e-beam evaporator (Daeki Hi-Tech Co., Ltd., Korea). Then, the mold with the target material and the micropatterned PDMS substrate were subjected to 10-s $O_2$ plasma treatment at a power of 100 W and an $O_2$ flow rate of 50 sccm to activate the material surface and selectively etch the PUA mold. The adhesion promoter, N-[3-(trimethoxysilyl)propyl]ethylenediamine, was spin-coated on the PDMS substrate at 5000 rpm, and the substrate was heated in a convection oven at 180 °C for 3 min to evaporate the solvent and activate the dangling bonds of the adhesion promoter. The plasma-treated PUA mold was then placed on pre-stretched PDMS by a customized linear stage and slightly pressed using a hand roller in the oven to facilitate the conformal contact between the mold and the substrate. After 4 min at 180 °C, the mold was slightly detached from the substrate, transferring the target material to the same, and the pre-strain was then slowly released.

### Configuration analysis and printing of various 3D nanostructures

Surface profiling and 3D imaging were performed using confocal laser scanning microscopy (VX-X1050, Keyence, Japan). Surface characteristics were also evaluated by field-emission scanning electron microscopy (Sirion, FEI, the Netherlands) and focused ion-beam scanning electron microscopy (Helios Nanolab, FEI, the Netherlands). All simulations were carried out using Abaqus CAE software (Dassault Systems, France). Experiments for analyzing the buckling configuration in Fig. 2 were carried out with an Au thickness of 100 nm. Experiments in Fig. 3a were carried out using a Pd thickness of 100 nm, and those in Fig. 3b were performed using a multilayer setup of 50 nm Au and 50 nm $SiO_2$.

## Electromechanical characterization of printed nanostructures and their evaluation as gas sensors

A customized linear stage and a commercial source meter (Keithley 2400, Keithley Instruments, USA) were used to measure electrical resistance under applied strain. In gas tests, 100-nm-thick Pd and $In_2O_3$ layers with a nano-serpentine pattern were fabricated and used as gas sensors. $H_2$ and $NO_2$ sensing tests were conducted using a customized gas chamber, and gas content was controlled using a mass flow controller. Target gases with the desired contents (0.2, 0.6, 1, and 1.4 vol% for $H_2$ and 1, 5, 10, and 20 ppm for $NO_2$) were prepared by controlling the flow rates of $H_2$, $NO_2$, $O_2$, and $N_2$ with similar ratios as the ambient air and supplying them to the gas chamber. During the gas test, the electrical resistance of the gas sensor was measured using a source meter (2635 B, Keithley Instruments, USA). In the gas selectivity test, the level of each tested gas was set close to the permissible exposure limit or the lower explosive limit according to the OSHA safety guidelines for toxic and hazardous substances 1915.1000[34]. During the gas tests of $In_2O_3$-based $NO_2$ sensor, visible light was illuminated by a commercialized illuminator (Lucky 7 LED, Seoul Semiconductor, Korea) with an output power of 100 W to facilitate the gas reaction.

## Data availability

The data that support the findings of this study are available from the corresponding author upon request.

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

## Acknowledgements

I.P. was supported by the National Research Foundation of Korea (NRF) grant funded by the Korean government (MSIT) [No. 2021R1A2C3008742]. I.P. was supported by the Technology Innovation Program (00144157, Development of Heterogeneous Multi-Sensor Micro-System Platform) funded by the Ministry of Trade, Industry &

Energy (MOTIE, Korea). S.J. and J.H.J were supported by the Institute of Information & Communications Technology Planning & Evaluation (IITP) grant funded by the Korean government (MSIT) (No. 2020-0-00914, Development of hologram printing downsizing technology based on holographic optical element(HOE)). S.J. and J.H.J. were supported by the Center for Advanced Meta-Materials (CAMM) funded by the Ministry of Science, ICT, and Future Planning as Global Frontier Project (CAMM No. 2014M3A6B3063707).

## Author contributions

J.A., J.H.J., and I.P. led the development of the concepts, designed the experiments, and interpreted the results. J.A. led the experimental work with support from J.J.H., Y.Je., Y.Ju., J.C., J.G., S.H.H., M.K., J.K., S.C., H.H., K.K., J.P., and S.J.

## Competing interests

The authors declare no competing interests.
