## [Peer Review File · Nature Communications]

Nanoscale Three-Dimensional Fabrication Based on Mechanically Guided AssemblyEVIEWER COMMENTS

Reviewer #1 (Remarks to the Author):

Manufacturing functional micro/nanostructures with complex 3D geometries has far-reaching implications across numerous engineering fields. Ahn et al. reported a nanoscale 3D manufacturing technique based on buckling-induced assembly in this manuscript. The authors presented a customized nano transfer strategy which applies to nanoribbons as narrow as 50 nm (Line 68), and demonstrated assembling a collection of line, mesh, and serpentine 2D geometries into 3D shapes. Experimental and computational/theoretical approaches were employed in this study. The authors also presented strain-insensitive gas sensors based on the 3D nanostructures. In general, this work represents an advance in the current research on manufacturing 3D structures at the nanoscale and these results may attract the attention of many other researchers and can inspire designs of more complex functional devices with novel application potentials. The manuscript can be considered for publication provided that the authors address some issues/inconsistencies listed below.

1) What did the authors mean by 4D printing?

The authors must define “printing” in the article, since the authors are trying to use a broader meaning than in general 3D or 4D printing. Merely being able to change the 3D shape reversibly by re-stretching the elastomer substrate is insufficient to be claimed as “4D.”

2) Line 18: The authors mentioned that “4D printing based on mechanically guided assembly” has been developed. The main advance of this article lies in expanding into the nanoscale.

By comparing this work to the previous publications on assembly by compressive buckling (for example, “Controlled buckling of semiconductor nanoribbons for stretchable electronics,” *Nature Nanotechnology* 1, 201, 2006; “Assembly of micro/nanomaterials into complex, three-dimensional architectures by compressive buckling,” *Science* 347, 15, 2015), some readers may think the novelty of this work is not sufficient. Please try to reinforce the originality of this proposed approach.

3) Is the narrowest 3D ribbon structure 50 nm as claimed in Line 68 or 200 nm as in Fig. 1c-i? I assume the narrow ribbons of 50 nm in Fig. 2e are only in 2D configuration.

What is the narrowest and/or thinnest 3D ribbon structure that is achievable by this approach?

4) The demonstrated geometries in this work all adopt rectangular overall 2D shapes. In Fig. 4b, the substrate pattern changes while the 2D structures are similar. Can the overall 2D geometry be arbitrary or customized, instead of only rectangular shapes?

5) Line 353: the authors mentioned using omnidirectional strain for more complex 3D nanostructures. As an easier goal, is this current approach directly applicable to bi-directional 3D manufacturing? Can the authors demonstrate a simple design, if possible?

6) There seem fracture locations on the 3D nano beams in the SEM in Fig. 1c.

Fig. 1c-i and Line 107: The 3D nano beams have cross-sectional dimensions of 200 nm by 80 nm. Given this aspect ratio and short longitudinal dimension, the strain level in the 3D beam after buckling should be higher than 3D thin films. Can the authors calculate the maximum strain/stress in the 3D nano beams and discuss whether it can lead to material fracture?

7) The authors used an analytical model from Ref. 20 to predict the 3D shapes. But the stretched configuration of the pillars on the elastomer substrate is no longer flat, as pointed out in Fig. 3c-ii. Fig. 3d-ii: the experiment result deviates from the analytical one, possibly due to that the pillars on the elastomer substrate is not flat after stretching. As the assumptions are different, the authors may not be able to directly adopt the formulae from Ref. 20. The authors should add a discussion.

Fig. S14: In the FEM, did the authors consider the elastomer substrate in simulations to accurately capture the deformed 3D shapes?

8) Fig. 3c-ii: Have the authors checked mesh convergence for the FEM study? Why did the authors choose to show von Mises stress contour for the substrate?

9) Line 199 and Fig. 3c-i: Is the critical pre-strain on the pillar 2.5% for controlling the buckling direction independent of ribbon dimensions (e.g., thickness and width) and materials?

Fig. 3e-i: Is the critical pillar width 3 μm for controlling the buckling mode independent of ribbon dimensions and materials? This value should depend on the stiffness difference between the 2D ribbon and the substrate pillar.

10) Fig. S16a: The electrical resistance curves for loading and unloading stages coincide perfectly. Repetition of the experiment is necessary. Additionally, will the curves for loading and unloading stay the same after a few cycles of stretching and releasing?

“Eelectrical” was misspelled.

11) The strain-insensitive concept may be no longer novel, as there are already many similar demonstrations.

Fig. 5: More repetitions of experiment to evaluate the gas sensing performance are necessary/required. How about the performance after many cycles of stretching/releasing? Will the behavior stay the same?

12) Fig. 5b and Line 311: The bound nanopattern H₂ gas sensor is neither sensitive nor stretchable. Can the authors elaborate why the bound nanopattern differs hugely from the 3D buckled nanopattern even without being stretched. Is it merely because the active surface area is approximately reduced by half (Line 306)?

More importantly, how does the performance of the proposed 3D gas sensors (buckled nanopatterns) compare to gold standard in gas sensing (e.g., commercial gas sensors) for both H₂ and NO₂?

13) Fig. 5e: It takes the NO₂ sensor a few hours to recover to its initial state after each measurement. In some cases, it could not return to its original electrical resistance. I am afraid this demonstration is not as exciting.

Very minor comments

14) The authors mentioned that the sensitive and stretchable gas sensor is for wearable applications. Wearable electronics usually have encapsulations to protect the components. Will the wearable gas sensor without encapsulation have other practical problems?

Line 323: The authors implied “various” gas sensors can be made. Can the authors discuss what strategies can be used for sensing different gases?

15) Line 340: mechanical “properties” of the PDMS substrate may not be the accurate word to describe the various strategies in designing the elastomer substrate in Fig. 3.

16) In Fig. S1, the inset pointing to Ref. 2 is from Ref. 12. Please double check.

17) Line 44: Metals and semiconductors are usually inorganic materials. The authors can change the words to be more rigorous.

18) Ref. 23 in the main text and Refs. 18 and 21 in the supplementary text do not have the universal citation format.

19) Formatting and typos

Fig. 1b: What did the authors mean by “mechanical stimulus?”

Fig. 2a: The font size may be too small to read.

Line 201: “<” should be “>.”

Line 309: “6b” should be “5b.”

Reviewer #3 (Remarks to the Author):

In this manuscript, the authors describe a new fabrication method to produce mechanically guided assembly of buckled nanostructures. Even though the pre-strained-substrate-induced buckling is not a brand-new concept, the reported fabrication method is novel and versatile and can enable deterministic buckling of nanoscale 2D patterns. The work is comprehensive and presented in a clear and logical way with detailed experimental procedures and extensive discussion. There are a few major strength and novelty of this work. (1) Using an innovative pattern transfer technique, this work extended the feature size limit of previously reported mechanically guided buckling systems. The technique appears to be relatively simply and reliable up to 10s of nanometers. (2) It is very important that the authors did thorough mechanical analysis and simulation to deterministically predict and reversely design the buckling direction and mode. This really broadens the scope of the work. (3) The materials chemistry is quite versatile, which would be crucial for potential applications. The authors proposed two approaches to transfer nano-materials with different adhesion strength. (4) The authors also demonstrated a proof-of-concept application of the buckled nano-patterns as strain-insensitive gas sensors.

Overall, the reviewer believes this manuscript is of the high quality of other papers on Nature Communications, if the following issues can be addressed. First and foremost, the use of the term “4D printing” is inaccurate and confusing to a general audience. The reviewer understands that there are a number of papers on 3D printing processes with some post-printing structural transformation claiming the concept of “4D printing”. Even though this terminology is not a consensus in the field, the reviewer

thinks it is acceptable if and only if (1) 3D printing is used to create 3D structures with a large degree of geometric freedom and (2) such 3D structure can be transformed after fabrication with an additional temporal degree of freedom (3D + 1D). However, in the context of this work, the authors used conventional nanofabrication techniques to create 2D patterns, transferred them to a pre-strained substrate, and then released the substrate to create 3D structures—this process is referred to in the manuscript as “4D printing”. The constant reference to 2D, 3D, and 4D appears to be quite confusing for readers. There are a lot of kirigami-inspired works that deform specific cut planar patterns in 3D (Choi, W. J. et al. *Nat. Mater.* 18, 820–826 (2019) for example). Should we consider them “4D printing” as well? The authors also refer to the pattern transferring process as printing, which is not accurate in a more modern sense of the word. To the reviewer’s understanding, printing (ink-jet printing in 2D or 3D printing) involves processes that can create/define patterns and structures on the fly. For example, an ink-jet printer can print any documents the user sends to it instead of just transferring/copying an existing document. Therefore, the reviewer would recommend reconsidering the use of “4D printing” and rephrasing the pattern transferring process “imprinting” or other more appropriate terms instead of “printing” as used in the manuscript.

Some minor areas for improvement are below. (1) It would be good to briefly explain what is Mode 1/2/3 buckling to general audience. (2) In Fig. 2e, there are some white dots on the edges of the ribbons. What is the cause? Are they related to O₂ plasmas treatment? Would H₂ plasma solve some of the oxidation issues without using protective coating? (3) A few sentences explaining the general principles of the electromechanical gas sensor would be helpful to readers. (4) More discussion on why the field needs to scale down the mechanically guided buckling to nm scale would make the motivation section stronger.

REVIEWER REPORT(S):

[Response to Reviewers #1' Comments]

General Comments to the Author:

Manufacturing functional micro/nanostructures with complex 3D geometries has far-reaching implications across numerous engineering fields. Ahn et al. reported a nanoscale 3D manufacturing technique based on buckling-induced assembly in this manuscript. The authors presented a customized nano transfer strategy which applies to nanoribbons as narrow as 50 nm (Line 68), and demonstrated assembling a collection of line, mesh, and serpentine 2D geometries into 3D shapes. Experimental and computational/theoretical approaches were employed in this study. The authors also presented strain-insensitive gas sensors based on the 3D nanostructures. In general, this work represents an advance in the current research on manufacturing 3D structures at the nanoscale and these results may attract the attention of many other researchers and can inspire designs of more complex functional devices with novel application potentials. The manuscript can be considered for publication provided that the authors address some issues/inconsistencies listed below.

Reply: We sincerely appreciate your careful review and valuable comments on our manuscript. We hope that the revised manuscript is suitable for publication in *Nature Communications*. The following text lists our point-by-point responses to each of the reviewer' comments, with the revised sentences presented in red.

Some specific comments:

[Comment 1] What did the authors mean by 4D printing? The authors must define “printing” in the article, since the authors are trying to use a broader meaning than in general 3D or 4D printing. Merely being able to change the 3D shape reversibly by re-stretching the elastomer substrate is insufficient to be claimed as “4D”.

Reply: We appreciate your insightful comments and agree that the term ‘4D’ is not accurate and can be confusing to the general audience. Initially, we used the term ‘4D printing’ because some recent papers reporting technologies based on mechanical assembly claimed to report 4D printings [R1, R2]. In these studies, authors may define 4D printing in a broad sense as ‘conventional printing + transformation over time (additional shape-morphing by external stimuli)’. However, after careful reconsideration, we concluded that the definition of ‘4D’ from reviewer #3 ‘(1) 3D printing is used to create 3D structures with a large degree of geometric freedom, and (2) such 3D structure can be transformed after fabrication with an additional temporal degree of freedom (3D + 1D).’ is more accurate and universal, and technologies based on mechanically guided assembly are a kind of the advanced 3D printings. Thus, we have revised ‘4D printing’ to ‘3D printing based on mechanically guided assembly’ in the overall manuscript including title as per the reviewer’s suggestion, and some representative changes are as follows (we attached a few examples here; all changes can be clearly seen in the

‘Manuscript_with_highlight’ file).

Title

Original text: “Nanoscale Four-Dimensional Printing Based on Mechanically Guided Assembly”

Revised text: “Nanoscale **Three-Dimensional Printing** Based on Mechanically Guided Assembly”

Introduction

Original text: “Given the difficulty of fabricating complex 3D structures using conventional two-dimensional (2D) planar processes such as inkjet/screen printing and lithography/ion-milling-based techniques, 3D and the recently developed four-dimensional (4D) printing methods have drawn much attention. In particular, 4D printing methods based on mechanically guided assembly (i.e., compressive buckling-based printing) allow the fabrication of complex 3D structures in thin and curvilinear forms and offer the advantages of wide material scope (e.g., metals, semiconductors, and polymers), high designability, precise controllability, scalability, and structural reversibility under strain, thus holding great promise for next-generation printing”

Revised text: “Given the difficulty of fabricating complex 3D structures using conventional two-dimensional (2D) planar processes such as inkjet/screen printing and lithography/ion-milling-based techniques, **recently developed and more advanced 3D printing methods** have drawn much attention. In particular, **3D printing methods based on mechanically guided assembly** (i.e., compressive buckling-based printing) allow the fabrication of complex 3D structures in thin and curvilinear forms and offer the advantages of wide material scope (e.g., metals, semiconductors, and polymers), high designability, precise controllability, scalability, and structural reversibility under strain, thus holding great promise for next-generation printing.”

Reference

- R1. Zhu, H. et al. Mechanically-Guided 4D Printing of Magneto-responsive Soft Materials across Different Length Scale. *Advanced Intelligent Systems* **4**, 3, 2100137 (2022).
- R2. Taylor, J. M. et al. Biomimetic and Biologically Compliant Soft Architectures via 3D and 4D Assembly Methods: A Perspective. *Advanced Materials* **34**, 16, 2108391 (2022).

[Comment 2] Line 18: The authors mentioned that “4D printing based on mechanically guided assembly” has been developed. The main advance of this article lies in expanding into the

nanoscale. By comparing this work to the previous publications on assembly by compressive buckling (for example, “Controlled buckling of semiconductor nanoribbons for stretchable electronics,” *Nature Nanotechnology* 1, 201, 2006; “Assembly of micro/nanomaterials into complex, three-dimensional architectures by compressive buckling,” *Science* 347, 15, 2015), some readers may think the novelty of this work is not sufficient. Please try to reinforce the originality of this proposed approach.

Reply: We appreciate your careful comment and agree that the discussion about novelty of this work should be clarified in the manuscript. As the reviewer discussed, the main novelty of this work is scaling-down of the 3D printing based on mechanically guided assembly to the nanoscale through the development of the nanotransfer printing compatible with elastomer substrate and configuration design method by the modulation of the substrate’s mechanical characteristics, which were impossible in the previous works. In addition to Abstract and Introduction of the main manuscript, we already described the details in Supplementary Notes 1 (i.e., summary and comparison of recently developed 3D printing based on mechanically guided assembly). However, we agree that some representative references pointed by the reviewer were missing in our manuscript. Therefore, we have added them in the related texts, and attached revised references for clear understanding of reviewer as follow.

Supplementary Notes 1

Related text: “**Summary of recently developed 3D printing based on mechanically guided assembly.** As discussed in the main text, 3D printing methods based on mechanically guided assembly have been actively studied to improve design diversity with various materials¹⁻⁹ and to develop new applications¹⁰⁻¹⁶ or inverse design techniques¹⁷. Although 3D nanostructures are in high demand¹⁸⁻²¹, **nanoscale fabrication is still challenging for most existing printing methods including mechanically guided assembly-based 3D printing**^{22,23}. Hence, we herein considered the printable size range as one of the core problems to be solved. To address this problem, we divided the 3D structures fabricated using mechanically guided assembly-based 3D printing into two parts, namely, bound and suspended sites, with a summary of recently reported printable sizes provided in Supplementary Fig. 1. It is worth noting that the longitudinal size of the printed structures was not considered for the following reasons. 1) For the suspended site, the length exceeds the width by one order of magnitude or more in most studies, as buckling easily occurs in high-aspect-ratio beams. 2) For the bound site, the related discussion is premature, as nanoscale alignment methods are yet to be developed, as discussed in the conclusion section. 3) Nevertheless, it is meaningful to reduce the width of the bound site as it is directly related to the design diversity and number of printable devices per unit area, and the suspended site shows width-dependent chemo-mechanical properties, as exemplified by nanowires with ultra-high surface area to volume ratios^{24,25}. As macro-/microscale printing does not suffer from adhesion-related problems (Supplementary Fig. 1), the sizes of the bound and suspended sites are similar, but the relative size of the bound site increases in the case of nanoscale printing because of the weak adhesion of the substrate (Supplementary Fig. 1). In addition, the difficulty of controlling the buckling configuration of the nanoscale beam has precluded the implementation of complex structures. Therefore, we herein developed a nanoscale transfer printing method on an elastomer substrate (Fig. 2) and controlled the configuration of the nanoscale buckled beam (Fig. 3) to realize configuration-designable nanoscale 3D printing and thus paved the way to universal 3D nanostructure printing.”

Introduction

Related text: “However, current techniques allowing 2D printing on stretchable elastomeric substrates suitable for 3D printing have size limitations, with stable adhesion guaranteed only for sizes above tens of micrometers because of low adhesion force at the nanoscale¹⁷. In addition, current configuration design method is based on the patterning of an adhesive layer that will become bound sites on the printing materials before transfer. However, applying this conventional method to extremely thin adhesives (i.e., self-assembled monolayers) and nanopatterned printing materials used in 2D nanotransfer printing is challenging. Therefore, the design and control of nanoscale buckling configuration (e.g., beam shape, direction, deflection, and mode) are much more difficult. Thus, the realization of configuration-designable nanoscale 3D printing based on mechanically guided assembly remains challenging, and the demand for 3D nanodevices for applications such as gas sensors, electrodes, thermoacoustic speakers, and optical devices remains unmet¹⁸ (Supplementary Notes 1).”

References

22. Xu, S. *et al.* Assembly of micro/nanomaterials into complex, three-dimensional architectures by compressive buckling. *Science* **347**, 154-159 (2015).
23. Sun, Y. A. *et al.* Controlled buckling of semiconductor nanoribbons for stretchable electronics. *Nat Nanotechnol* **1**, 201–207 (2006).

[Comment 3] Is the narrowest 3D ribbon structure 50 nm as claimed in Line 68 or 200 nm as in Fig. 1c-i? I assume the narrow ribbons of 50 nm in Fig. 2e are only in 2D configuration. What is the narrowest and/or thinnest 3D ribbon structure that is achievable by this approach?

Reply: We appreciate your careful comment. In our study, the narrowest nanoribbons were demonstrated down to the width of 50 nm in 2D and down to 200 nm in 3D because there is a limitation of the stable printing conditions. As the reviewer suggested, we conducted additional experiments for finding the narrowest dimension in nanoscale 3D printing. Because our Si master mold with the width of 50 nm does not satisfy the requirement for the 3D printing (it is too shallow for e-beam deposition of the target material with a proper thickness), we fabricated 3D nanoribbon with a width of 100 nm. The results show stable 3D nanoribbon array as attached in Supplementary Fig. 2, and thus, we expect that the developed process may be applicable for the 3D nanoribbon with the width of 50 nm if a proper Si master mold is supplied. However, one remaining issue needs to be resolved in the subsequent research. As the width of the 3D nanoribbon decreases, the directional preference of the buckling also decreases, and thus, the nanoribbons tend to be entangled as shown in Supplementary Fig. 2. In detail, the nanoribbon fabricated by e-beam deposition requires at least a thickness of 40–60 nm to maintain its original shape even after being transferred. If the width of the ribbon becomes similar to its thickness (i.e., it becomes a rectangular nanowire, not a thin nanoribbon), it can be buckled into the either vertical or horizontal direction because there is no difference in the second moment of area of the cross-section between each direction. Therefore, to realize the perfectly controlled 3D nanostructures with narrower width, in-depth follow-up research to control the buckling direction in both vertical and horizontal directions should be conducted. In conclusion, as the reviewer discussed, we have conducted the additional experiments for

demonstrating 3D nanoribbon as narrow as possible, and added them to the Supplementary Fig. 2d as follows.

Supplementary Fig. 2d

Original Fig.:

Supplementary Fig. 2: Microscopic characterization of the substrate, target material, mold, and a representative nanoline array. **a**, Confocal laser scanning microscopy and optical microscopy images of a 10-μm-thick micropatterned polydimethylsiloxane substrate. **b**, Top-view scanning electron microscopy image of the target material (Au) on a nanopatterned poly(urethane acrylate) mold with a linewidth of 800 nm. **c**, Scanning electron microscopy images of a buckled nanoline array with a linewidth of 800 nm and a thickness of 100 nm acquired at various angles of view.

Revised Fig.:

[Comment 4] The demonstrated geometries in this work all adopt rectangular overall 2D shapes. In Fig. 4b, the substrate pattern changes while the 2D structures are similar. Can the overall 2D geometry be arbitrary or customized, instead of only rectangular shapes?

Reply: We appreciate your valuable comment and agree that the demonstrated 2D nanopatterns are relatively simple compared with the diverse substrate patterns. The developed nanotransfer printing method is hardly affected by the shape of the nanopattern, and thus, various 2D nanopatterns can be transferred. In addition to the nanopatterns previously shown in Fig. 4a (line, mesh, and serpentine pattern) and in Supplementary Fig. 8a (dot pattern), we have demonstrated two more complex patterns (i.e., arbitrary patterns and customized cross pattern) as the reviewer suggested, and added them to Supplementary Fig. 8a as follows.

Supplementary Fig. 8a

Original Fig.:

Supplementary Fig. 8: Scanning electron microscopy images showing the effects of (a) pattern shape.

Revised Fig.:

Supplementary Fig. 8: Scanning electron microscopy images showing the effects of (a) pattern shape: (i) dot, (ii) mesh, (iii) arbitrary, and (iv) cross patterns.

[Comment 5] Line 353: the authors mentioned using omnidirectional strain for more complex 3D nanostructures. As an easier goal, is this current approach directly applicable to bi-directional 3D manufacturing? Can the authors demonstrate a simple design, if possible?

Reply: We are thankful to the reviewer for the valuable comment. Theoretically, it seems that the developed printing process can be applied for the omnidirectional 3D nanofabrication because the nanotransfer printing on the pre-stretched elastomer substrate is not a strain-dependent technique. However, in our trial for showing feasibility as per the reviewer's suggestion, we failed to fabricate the omnidirectional complex 3D nanostructures because of the following two technical issues: 1) More precise alignment is required to locate the 2D nanopatterns on the 2D micro/nanopattern of the substrates and control the final 3D structure. To handle this issue, methods for at least a few microns-scale alignments during nanotransfer printing need to be developed. 2) Because of the complex omnidirectional strain and misalignment, the suspended nanostructures were broken or deformed into the unwanted irregular shapes. Therefore, we could conclude that although nanoscale 3D printing with the omnidirectional strain has a great potential to realize more complex 3D nanostructures with high degree of freedom, it is needed to be studied in more detail based on the background of the current work in a subsequent research.

[Comment 6] There seem fracture locations on the 3D nano beams in the SEM in Fig. 1c. Fig. 1c-i and Line 107: The 3D nano beams have cross-sectional dimensions of 200 nm by 80 nm. Given this aspect ratio and short longitudinal dimension, the strain level in the 3D beam after buckling should be higher than 3D thin films. Can the authors calculate the maximum strain/stress in the 3D nano beams and discuss whether it can lead to material fracture?

Reply: We appreciate your valuable comments. The fracture of the 3D nanobeams observed in Fig. 1c. occurred when the sample was cut by the ion-milling process to obtain the side-view image. As show in Supplementary Fig. 2c., the original large-area sample has almost zero fracture defect. In the case of stress analysis, we would like to explain with the following Euler's buckling theory [R1].

Critical stress required for beam buckling is calculated using the equation ' $\sigma_{crit} = \frac{\pi^2 E}{(\frac{L}{r})^2}$ ', where σ_{crit} means critical stress required for buckling, E means Young's modulus of material, L means effective length of the beam, and r means least radius of gyration. Here, r is defined as $\sqrt{\frac{I}{A}}$, where I means the second moment of area of the cross-section and A means the area of the cross-section of the beam.

Therefore, in our study, r can be expressed as $\sqrt{\frac{h^2}{12}}$ because I is calculated as $\frac{bh^3}{12}$ and A is calculated as bh , where b is width of the beam and h is thickness of the beam. It means that there is no theoretical difference in critical stress required for buckling between beam and thin film. In addition, according to these equations and the Fig. R1, material yield or fracture occurs when the buckling critical load (σ_{crit}) is higher than yield stress (σ_Y) or fracture stress. For

example, in Fig 1c., σ_{crit} is calculated as 17.5 MPa (where $E = 79 \text{ GPa}$, $b = 200 \text{ nm}$, $h = 80 \text{ nm}$, $L = 25 \mu\text{m}$) and σ_Y is known as 128-758 MPa. Therefore, there will be no yield or fracture in our 3D nanobeam because σ_{crit} is much lower than σ_Y . The related figures are attached below for clearer understanding.

Supplementary Fig. 2: Various optical and scanning electron microscopic (SEM) images of the substrate, target material, and nanoline array. a, Confocal laser scanning microscopy and optical microscopy images of a 10- μm -thick micropatterned polydimethylsiloxane substrate. b, Top-view scanning electron microscopy image of the target material (Au) on a nanopatterned poly(urethane acrylate) mold with a linewidth of 800 nm. c, SEM images of a buckled nanoline array with a linewidth of 800 nm and a thickness of 100 nm acquired at various angles of view. d, Top- and side-view SEM images of a buckled nanoline array with a linewidth of 100 nm and a thickness of 60 nm.

Fig. R1. Schematic graph of critical stress vs slenderness ratio.

Reference

[R1]. Bhoi, R. *et al.* Study of buckling behaviour of beam and column subjected to axial loading for various rolled I sections. *International Journal of Innovative Research in Science, Engineering and Technology* **3**, 17639–17645 (2014).

[Comment 7] The authors used an analytical model from Ref. 20 to predict the 3D shapes. But the stretched configuration of the pillars on the elastomer substrate is no longer flat, as pointed out in Fig. 3c-ii. Fig. 3d-ii: the experiment result deviates from the analytical one, possibly due to that the pillars on the elastomer substrate is not flat after stretching. As the assumptions are different, the authors may not be able to directly adopt the formulae from Ref. 20. The authors should add a discussion. Fig. S14: In the FEM, did the authors consider the elastomer substrate in simulations to accurately capture the deformed 3D shapes?

Reply: We appreciate your insightful comment. As you mentioned, for the above analysis (analytical model and simulation in Supplementary Fig. 20), the top surface of the pillar was assumed as flat under the applied strain. The assumption for the flat surface of the pillar is reasonable in this analysis for the following reason. As shown in the added figure below (Supplementary Fig. 16), the amount of the edge deformation and the deformed region under the strain is relatively negligible compared to the pillar width (originated from the relatively small ϵ_{pillar}). Specifically, even under the applied pre-strain of 30% which is the maximum strain applied in this study, the maximum deformation in z-direction is 1.5 μm , and the effective non-flat region of the top surface of the pillar (i.e., region where the maximum deformation in z-direction is larger than 2% of the pillar width) is 2 μm , while the width of the pillar is approximately 50 μm . Furthermore, if the overall top surface of the pillar was not flat, 2D nanopatterns might not be transferred onto the pillar surface because we did not apply any pressure except for the weight of PUA film for the conformal contact on the mold during the nanotransfer process. Therefore, the top surface of the pillar was assumed as flat and the equation from Ref. 20 was adopted, although there is minor deformation in the edge of the pillar under the applied strain. The deviation between experimental results, FEM simulation, and analytical method in Fig. 3d-ii may be originated from the experimental errors such as

imprecisely applied strain or thickness variation of the elastomer substrate. In conclusion, because of the small amount of the edge deformation in the tiny region, it affected the buckling direction only, not the overall deflection configuration. However, as the reviewer suggested, we agree that the related discussion should be added for clearer understanding. Therefore, we have added the additional figure and texts as follow.

Results and discussion

Original text: “As shown in Fig. 3d-ii and Supplementary Fig. 14, the experimental results regarding the maximum beam deflection as a function of ϵ_{trench} (in this research, the trench strain is same as the pre-strain (ϵ_{pre}) in Equation (3)) were well-matched with the results of FEM simulation and analytical calculations. In addition, as shown in Fig. 3d-iii, the configuration of the single beam (measured using CLSM (Supplementary Fig. 15)) was well-matched with the results of the FEM simulation.”

→

Revised text: “As shown in Fig. 3d-ii and Supplementary Fig. 14, the experimental results regarding the maximum beam deflection as a function of ϵ_{trench} (in this research, the trench strain is same as the pre-strain (ϵ_{pre}) in Equation (3)) were well-matched with the results of FEM simulation and analytical calculations. In addition, as shown in Fig. 3d-iii, the configuration of the single beam (measured using CLSM (Supplementary Fig. 15)) was well-matched with the results of the FEM simulation. **Here, for the above analysis, the top surface of the pillar was assumed as flat under the applied strain. Because the amount of the edge deformation under the strain is relatively minor compared to the pillar width (originated from the relatively small ϵ_{pillar}) (Supplementary Fig. 16), it affected the buckling direction only, not the overall deflection configuration.”**

Added Supplementary Fig.:

[Comment 8] Fig. 3c-ii: Have the authors checked mesh convergence for the FEM study? Why did the authors choose to show von Mises stress contour for the substrate?

Reply: We appreciate your careful comments. First, we have checked the mesh convergence for the FEM simulation as the reviewer suggested. As shown in Fig. R2, the current mesh state in Fig. 3c-ii is enough for verifying the edge effect, although it is not highly dense because we used a simplified model just to observe the edge deformation of the patterned substrate. Second, we demonstrated the substrate deformation with von Mises stress contour in Fig. 3c-ii to show the transferred strain energy from the substrate. Because the strain energy caused by the applied strain at each end of the substrate is mainly distributed on the trench and the center of the pillar except the edge of the pillar, the edge looks deformed upward, and thus, buckling occurs in the upward direction when 3D printing is performed at large pre-strain. We believe that this simulation information including von Mises stress contour may help the readers to more intuitively understand the edge effect on the buckling direction.

Fig. R2. Maximum stress depending on the number of elements in mesh in FEM simulation.

[Comment 9] Line 199 and Fig. 3c-i: Is the critical pre-strain on the pillar 2.5% for controlling the buckling direction independent of ribbon dimensions (e.g., thickness and width) and materials? Fig. 3e-i: Is the critical pillar width 3 μm for controlling the buckling mode independent of ribbon dimensions and materials? This value should depend on the stiffness difference between the 2D ribbon and the substrate pillar.

Reply: We appreciate your insightful comments. The threshold values (e.g., critical pre-strain or critical pillar dimensions) for controlling the buckling configurations (e.g., direction and mode) can be changed depending on the material and 2D structural characteristics because the interactive force between substrate and the dimension can be changed when the buckling occurs. As discussed in our response to comment #6 and shown in the corresponding equation, the critical stress for generating buckling is calculated as $\sigma_{crit} = \frac{\pi^2 E}{(\frac{L}{r})^2} = \frac{\pi^2 E}{(\frac{L}{\frac{h}{\sqrt{12}}})^2}$, in this study. Here,

although width of the nanoribbon does not affect the critical stress, material thickness, suspended length, and Young's modulus of material are directly related with the critical stress. Therefore, changing the dimension or material may generate the different critical stress for buckling. If the critical stress for buckling is changed, the required adhesion force between material and substrate and the applied stress on pillar will be also changed. These variations can not only cause the change of threshold values for controlling the buckling configuration but also increase possibility of the slip of the nanoribbon or bending of the pillar, as the reviewer discussed. However, considering the nonlinear dynamics of the hyperelastic substrate, it is still challenging to calculate the threshold values with all of the abovementioned parameters using analytical or numerical methods. Thus, we analyzed the detailed threshold values through empirical methods based on the analytical background as suggested in our study. Because it is a time-consuming to conduct the global experiments with all different parameters, we conducted the local analysis using a couple of representative materials such as Au, SiO₂, and In₂O₃ with the thickness of 80 nm and 100 nm and the width of 200 nm and 800nm, and the results showed almost the same threshold values for each material as shown in Fig. 3. This is because there is already big difference (i.e., more than a few orders) between the Young's modulus of the nanoribbon (e.g., Au= 76 GPa, SiO₂= 74 GPa, and In₂O₃= 145 GPa) and the substrate (PDMS= 870 KPa) and between the thickness of the nanoribbon (100 nm) and substrate (10 μm) compared with the difference between materials or dimensions. Therefore,

similar threshold values are obtained as the overall change on the interaction between substrate and material is relatively negligible. We thank the reviewer again for the insightful feedback.

[Comment 10] Fig. S16a: The electrical resistance curves for loading and unloading stages coincide perfectly. Repetition of the experiment is necessary. Additionally, will the curves for loading and unloading stay the same after a few cycles of stretching and releasing? “Eelectrical” was misspelled.

Reply: We appreciate your valuable comment and agree that the cyclic test for the electrical resistance change depending on the applied strain should be conducted and discussed in the manuscript. For accurate analysis, we used liquid metal electrodes that shows negligible electrical resistance change under the applied strain and four points method that can remove the contact resistance effect to remove all potential effects from the electrodes. As shown in Supplementary Fig. 17, although the electrical resistance curve as a function of the applied strain slightly varies depending on each cycle, the overall resistance change upon the applied 34% trench strain is less than 3% and the loading/unloading curve shows almost zero hysteresis. As the suspended (i.e., buckled) site can absorb most of the strain, the viscoelastic behavior of the substrate did not affect the nanostructure, and thus, no electrical or mechanical hysteresis was observed upon strain variation. Thus, we have made modifications (including correcting typos) as per the reviewer’s suggestion and added them to supplementary Fig. 17 as follows.

Supplementary Fig. 17

Original Fig.:

Supplementary Fig. 17: Electromechanical characterization of the buckled three-dimensional nanostructure. a, Maximum deflection and relative resistance as functions of applied strain. **b,** Confocal laser scanning microscopy images of the buckled three-dimensional nanostructure acquired at different applied strains. As the suspended (i.e., buckled) site can absorb most of the strain, the viscoelastic behavior of the substrate did not affect the nanostructure, and no electrical or mechanical hysteresis was observed upon strain variation.

Revised Fig.:

[Comment 11] The strain-insensitive concept may be no longer novel, as there are already many similar demonstrations. Fig. 5: More repetitions of experiment to evaluate the gas sensing performance are necessary/required. How about the performance after many cycles of stretching/releasing? Will the behavior stay the same?

Reply: We appreciate your insightful comment and agree that the gas sensing performance after the cyclic test with the repeated stretching/releasing should be discussed in the manuscript. Thus, we have evaluated the sensing performance before/after the cyclic test, and the result shows that the fabricated nano-serpentine Pd maintained its H₂ sensing performance even after the cyclic test, demonstrating similar response against H₂ gas as shown in Supplementary Fig. 18. Although the overall sensitivity looks slightly decreased as the number of the cycles increase, it may be caused by the material's gas sensing characteristic or systematic stabilization process which means the sensing performance can vary slightly during the cyclic test with the repeated exposure to the air/H₂ gas as reported in recent papers [R1-R3]. These issues are not related with the mechanical strain applied in our test and it can be solved by adopting Pd-based alloy or composite with better gas sensing stability. Therefore, we could conclude that the fabricated sensor is not damaged during the cyclic test of the repeated stretching/releasing, and the variation of 11.7% (i.e., difference between sensor responses to the H₂ gas during the stretching/releasing cyclic test) is still within the original sensor variation of 12.0% (i.e., difference between sensor responses to the H₂ gas during the repeated exposure to the air/H₂ gas) mentioned in the manuscript. As the reviewer suggested, the above result

should be added for clear understanding of the readers in our manuscript. Therefore, we have added the additional figure and texts as follows.

Results and discussion

Original text: “Figure 5b shows the gas sensing performances of buckled and bound nanopatterns under identical conditions. When both sensors were exposed to H₂, the buckled nanopattern showed 27- and 20-fold higher responses than the bound nanopattern at H₂ levels of 1 and 1.4 vol%, respectively. Notably, the buckled nanopattern maintained its response to 1 vol% H₂ with a variance of 12.0% when an external strain of up to 30% was sequentially applied, while the bound nanopattern was broken under the action of external strain of 7.5%. Figure 5c presents the selectivity of the buckled nanopattern–based sensor.”

→

Revised text: “Figure 5b shows the gas sensing performances of buckled and bound nanopatterns under identical conditions. When both sensors were exposed to H₂, the buckled nanopattern showed 27- and 20-fold higher responses than the bound nanopattern at H₂ levels of 1 and 1.4 vol%, respectively. Notably, the buckled nanopattern maintained its response to 1 vol% H₂ with a variance of 12.0% when an external strain of up to 30% was sequentially applied, while the bound nanopattern was broken under the action of external strain of 7.5%. **In addition, the experiments for sensing performance after a cyclic test with the repeated loaded/unloaded strain were conducted as shown in Supplementary Fig.18. The fabricated sensor was not damaged during the cyclic test of the repeated stretching/releasing, and the variation of 11.7% (i.e., difference between sensor responses to the H₂ gas during the stretching/releasing cyclic test) is still within the original sensor variation of 12.0% (i.e., difference between sensor responses to the H₂ gas during the repeated exposure to the air/H₂ gas). Figure 5c presents the selectivity of the buckled nanopattern–based sensor.”**

Added Supplementary Fig.:

Supplementary Fig. 18: Relative resistance of a Pd nano-serpentine pattern with buckling depending on the H₂ level, external strain, and number of the repeated loaded/unloaded strain of 30%.

Reference

[R1] Zhao, Z.-J., *et al.* 3D layer-by-layer Pd-containing nanocomposite platforms for enhancing the performance of hydrogen sensors. *ACS sensors* **5**, 2367-2237 (2020).

[R2] Darmadi, I. *et al.* High-performance nanostructured palladium-based hydrogen sensors—current limitations and strategies for their mitigation. *ACS sensors* **5**, 3306-3327 (2020).

[R3] Gao, M., *et al.* Palladium-decorated silicon nanomesh fabricated by nanosphere lithography for high performance, room temperature hydrogen sensing. *Small* **14**, 1703691 (2018).

[Comment 12] Fig. 5b and Line 311: The bound nanopattern H₂ gas sensor is neither sensitive nor stretchable. Can the authors elaborate why the bound nanopattern differs hugely from the 3D buckled nanopattern even without being stretched. Is it merely because the active surface area is approximately reduced by half (Line 306)? More importantly, how does the performance of the proposed 3D gas sensors (buckled nanopatterns) compare to gold standard in gas sensing (e.g., commercial gas sensors) for both H₂ and NO₂?

Reply: We appreciate your insightful comment. First, there are two main reasons why the suspended nanopattern showed much higher sensing performance compared with the bound nanopattern: 1) suspended nanopattern has approximately twice the active surface area to volume ratio of the bound nanopattern, as the reviewer discussed. 2) H₂ absorption reaction of Pd with the bound nanopattern is restricted by the clamping effect of substrate [30]. According to the literature, when Pd reacts with H₂ gas and changed to PdH_x, its volume expands up to 10%. If the Pd is bound to the substrate and the pattern thickness of Pd is thin enough to be affected by the substrate, Pd absorbs H₂ much less than its original capacity. We already described this in the manuscript and the related texts are attached below for your reference.

Second, as the reviewer said, it is necessary to compare the developed gas sensor with the commercialized sensors, however, it is still premature for the stretchable gas sensors. In this study, we focused on showing the merits of our structure and demonstrating the feasibility of the stretchable gas sensor, not optimizing the sensing performance (e.g., sensitivity and response time). Thus, the demonstrated performance is just comparable with the recently reported rigid gas sensors (e.g., for H₂ gas sensor at H₂ gas 1%, sensitivity of the current work: 9% and sensitivity of the recent work: 2% [30] and for NO₂ gas sensor at NO₂ 10 ppm, sensitivity of the current work: 55% and sensitivity of the recent work: 110% [R1]). The performance of the developed sensor must be improved further in subsequent research, and it will be possible with many well-known methods such as catalysts coating and post processing, which will be discussed in detail in reply to comment 13. However, in terms of performance for stretchability, the developed sensor shows superior merits compared with the previous sensors. There is no standardized stretchable gas sensor in the industrial field, and existing small number of reported papers handling stretchable gas sensor show a large response change under strain input (e.g., for NO₂ gas sensor under 10 % strain, variance of the current work:

12.5% and variance of the recent work: 250% [29]). Thus, we can conclude that our sensor still has novelty in terms of excellent strain-independent characteristics with the comparable sensing performance.

Results and discussion

Related text: “Subsequently, the 3D nanostructure based on the nano-serpentine pattern was used as a Pd-based H₂ sensor, offering three advantages over conventional bound nanopatterns in view of its unique buckled structure. Specifically, the utilization of all four sides of the nanoline pattern increased the active surface area, the effect of constraining the volume expansion of Pd by the bound substrate during the reaction was eliminated³⁰, and strain-insensitive sensing abilities were achieved as mentioned above.”

References

29. Yi, N. *et al.* Stretchable gas sensors for detecting biomarkers from humans and exposed environments. *TrAC - Trends in Analytical Chemistry* **133**, 116085 (2020).
30. Zhao, Z. J. *et al.* Wafer-scale, highly uniform, and well-arrayed suspended nanostructures for enhancing the performance of electronic devices. *Nanoscale* **14**, 1136–1143 (2022).
- R1. Ma, D. *et al.* Highly sensitive room-temperature NO₂ gas sensors based on three-dimensional multiwalled carbon nanotube networks on SiO₂ nanospheres. *ACS Sustainable Chemistry & Engineering* **8**, 13915-13923 (2020).

[Comment 13] Fig. 5e: It takes the NO₂ sensor a few hours to recover to its initial state after each measurement. In some cases, it could not return to its original electrical resistance. I am afraid this demonstration is not as exciting.

Reply: We appreciate your insightful comment. As you discussed, the demonstrated NO₂ gas sensor has a signal drift and relatively slow response with the rise time of tens of minutes and the fall time of a few hours. However, this issue was ascribed from the material characteristics itself, not from our printing process. It is well known that most of the metal oxide gas sensors, especially activated by light sources, have slow response and signal drift. Thus, many strategies have been developed to solve them, such as coating catalysts, long-time stabilization, thermal treatment, optimization of power source, or signal processing [R1, R2]. In this study, we just demonstrated a proof-of-concept application of the buckled nanopatterns as strain-insensitive gas sensors without any post processing because we concluded that it is enough for showing our novelty and applicability without confusion. However, we agree that studying these issues can be an interesting follow-up research, and thus, we have added an example of signal processing for clear understanding. By calculating the time derivative of the original sensor responses, the gas concentration could be quickly evaluated by the value of time derivative with dramatically reduced rise and fall times down to a few seconds and a few minutes, respectively.

Fig. R3. a, Fig. 5e: relative resistance of In₂O₃ nano-serpentine pattern with buckling as a function of NO₂ level. **b**, time derivative of sensor response in Fig. 5e.

References

- R1. Cho, I. *et al.* Monolithic micro light-emitting diode/metal oxide nanowire gas sensor with microwatt-level power consumption. *ACS sensors* **5**, 563-570 (2020).
- R2. Kumar, R. *et al.* Room-temperature gas sensors under photoactivation: from metal oxides to 2D materials. *Nano-Micro Letters* **12**, 1-37 (2020).

[Comment 14] The authors mentioned that the sensitive and stretchable gas sensor is for wearable applications. Wearable electronics usually have encapsulations to protect the components. Will the wearable gas sensor without encapsulation have other practical problems? Line 323: The authors implied “various” gas sensors can be made. Can the authors discuss what strategies can be used for sensing different gases?

Reply: We appreciate your insightful comments. First, although we just demonstrated the feasibility of sensitive and stretchable gas sensor using the developed 3D printing without encapsulation, as the reviewer mentioned, general wearable electronics as final products require proper encapsulation. In the case of gas sensor, if there is no encapsulation layer, the mechanical impacts or dust particles in the atmosphere can degrade the gas sensors. Therefore, encapsulation with soft and only-gas-permeable materials such as microporous elastomer will be helpful to protect the stretchable gas sensor from damage during daily use.

Second, current research trend to improve the selectivity of the gas sensors for various gases is focused on development of array of multiple gas sensors showing different responses to several gases and adoption of machine learning to distinguish the species and concentration of each gas using the measured multiple data [R1]. Therefore, one of the main issues in sensing different gases is how to fabricate and integrate multiple types of gas sensors with a simple process. In terms of material and structural diversity, the developed 3D printing based on mechanically guided assembly utilizes physical vapor deposition process, and most metal and metal oxide can be printed as shown in Fig. 2. It means that the previously studied strategies based on material and structural design for fabricating various gas sensors, such as applying diverse semiconductor metal oxides [R1], adding noble metal catalysts [R2], and fabricating heterogeneous structures for heterojunction effects [R3], can be easily realized using the developed 3D printing process. Therefore, we concluded that the developed 3D printing may

efficiently enable the application of previous strategies to detect different gases even for stretchable gas sensors with suspended and nanostructured form.

References

- R1. Kang, M. *et al.* High Accuracy Real-Time Multi-Gas Identification by a Batch-Uniform Gas Sensor Array and Deep Learning Algorithm. *ACS sensors* **7**, 430-440 (2022).
- R2. Del, D. *et al.* Pt Nanostructures Fabricated by Local Hydrothermal Synthesis for Low-Power Catalytic-Combustion Hydrogen Sensors. *ACS Applied Nano Materials* **4**, 7-12 (2020).
- R3. Liu, Y. *et al.* Chemiresistive gas sensors based on hollow heterojunction: a review. *Advanced Materials Interfaces* **8**, 2002122 (2021).

[Comment 15] Line 340: mechanical “properties” of the PDMS substrate may not be the accurate word to describe the various strategies in designing the elastomer substrate in Fig. 3.

Reply: We appreciate your valuable comment and agree that “mechanical properties” is not a proper expression for explaining the developed strategies. Thus, we have replaced the word “mechanical properties” to “mechanical characteristics”, and revised the appropriate section as follows.

Abstract

Original text: “Herein, a configuration-designable nanoscale 4D printing is suggested through a robust nanotransfer methodology and design of substrate’s mechanical properties. Covalent bonding–based two-dimensional nanotransfer allowing for nanostructure printing on elastomer substrates is used to address fabrication problems, while the feasibility of configuration design through the modulation of substrate’s mechanical properties is examined using analytical calculations and numerical simulations, allowing printing of various 3D nanostructures.”

Revised text: “Herein, a configuration-designable nanoscale 4D printing is suggested through a robust nanotransfer methodology and design of substrate’s mechanical **characteristics**. Covalent bonding–based two-dimensional nanotransfer allowing for nanostructure printing on elastomer substrates is used to address fabrication problems, while the feasibility of configuration design through the modulation of substrate’s mechanical **characteristics** is examined using analytical calculations and numerical simulations, allowing printing of various 3D nanostructures.”

Introduction

Original text: “Regarding the design part, micropatterning is used to modulate elastomer’s

mechanical properties and thus enable the rational design and prediction of the printed buckling configuration (i.e., direction, deflection, and mode) for a given 2D pattern.”

Revised text: “Regarding the design part, micropatterning is used to modulate elastomer’s mechanical **characteristics** and thus enable the rational design and prediction of the printed buckling configuration (i.e., direction, deflection, and mode) for a given 2D pattern.”

Results and discussion

Original text: “Herein, we suggest that the abovementioned parameters can be controlled by designing the mechanical properties of the micropatterned substrate, e.g., surface strain and Poisson’s effect.”

“Subsequently, a method for controlling the buckling configuration (i.e., direction, deflection, and mode) through the design of substrate’s mechanical properties was developed.”

Revised text: “Herein, we suggest that the abovementioned parameters can be controlled by designing the mechanical **characteristics** of the micropatterned substrate, e.g., surface strain and Poisson’s effect.”

“Subsequently, a method for controlling the buckling configuration (i.e., direction, deflection, and mode) through the design of substrate’s mechanical **characteristics** was developed.”

Conclusion

Original text: “Nanotransfer to the elastomer substrate was achieved by modulating the adhesion force between the polymer mold, target material, and the elastomer substrate, and the relationship between substrate’s mechanical properties and the final buckling configuration was studied to achieve rational design of fabricated 3D nanostructures.”

Revised text: “Nanotransfer to the elastomer substrate was achieved by modulating the adhesion force between the polymer mold, target material, and the elastomer substrate, and the relationship between substrate’s mechanical **characteristics** and the final buckling configuration was studied to achieve rational design of fabricated 3D nanostructures.”

[Comment 16] In Fig. S1, the inset pointing to Ref. 2 is from Ref. 12. Please double check.

Reply: We appreciate your careful comment and apologize for our mistake. We have re-checked all references in Supplementary Fig. 1, and have made modifications as per the reviewer’s suggestion as follows.

Supplementary Information

Original supplementary Fig. 1:

Supplementary Fig. 1: Summary of recently reported three-dimensional printing methods based on mechanically guided assembly.

Revised supplementary Fig. 1:

[Comment 17] Line 44: Metals and semiconductors are usually inorganic materials. The authors can change the words to be more rigorous.

Reply: We appreciate your valuable comment and agree that current expression in line 44 “e.g., metals, semiconductors, and inorganic materials” is not accurate. Thus, we have made modifications as per the reviewer’s suggestion in Introduction as follows.

Introduction

Original text: “In particular, 4D printing methods based on mechanically guided assembly (i.e., compressive buckling–based printing) allow the fabrication of complex 3D structures in thin and curvilinear forms and offer the advantages of wide material scope (e.g., metals, semiconductors, and inorganic materials), high designability, precise controllability, scalability, and structural reversibility under strain, thus holding great promise for next-generation printing.”

Revised text: “In particular, 3D printing methods based on mechanically guided assembly

(i.e., compressive buckling–based printing) allow the fabrication of complex 3D structures in thin and curvilinear forms and offer the advantages of wide material scope (e.g., metals, ceramics, and polymers), high designability, precise controllability, scalability, and structural reversibility under strain, thus holding great promise for next-generation printing”

[Comment 18] Ref. 23 in the main text and Refs. 18 and 21 in the supplementary text do not have the universal citation format.

Reply: We appreciate your valuable comment and apologize for our mistakes. We have made modifications in the reference format as per the reviewer’s suggestion as follows.

References

Original text: “24. Adv Healthcare Materials - 2021 - Jeong - Ultra-Wide Range Pressure Sensor Based on a Microstructured Conductive.pdf.”

“18. Collectively Exhaustive Hybrid Triboelectric Nanogenerator Based on Flow-Induced.pdf.”

“21. Advanced Energy Materials - 2022 - Ahn - All-Recyclable Triboelectric Nanogenerator for Sustainable Ocean Monitoring.pdf.”

→

Revised text: “24. Jeong, Y. *et al.* Ultra-Wide Range Pressure Sensor Based on a Microstructured Conductive Nanocomposite for Wearable Workout Monitoring. *Advanced Healthcare Materials* **10**, 2001461 (2021).”

“18. Kim, J.-S. *et al.* Collectively Exhaustive Hybrid Triboelectric Nanogenerator Based on Flow-Induced Impacting-Sliding Cylinder for Ocean Energy Harvesting. *Advanced Energy Materials* **12**, 2103076 (2022).”

“21. Ahn, J. *et al.* All-Recyclable Triboelectric Nanogenerator for Sustainable Ocean Monitoring Systems. *Advanced Energy Materials* **12**, 2201341 (2022).”

[Comment 19] Formatting and typos.

Fig. 1b: What did the authors mean by “mechanical stimulus?”

Fig. 2a: The font size may be too small to read.

Line 201: “<” should be “>.”

Line 309: “6b” should be “5b.”

Reply: In the case of ‘mechanical stimulus’, it originally means broad expression for the applied mechanical tensile strain on the PDMS substrate (i.e., substrate’s pre-strain). However, we agree that this broad expression can cause confusion. Therefore, we have made modifications as per the reviewer’s suggestion as follows.

Results and discussion

Original fig. 1:

Fig. 1: Concept and realization of configuration-designable nanoscale three-dimensional printing.

→

Revised fig. 1:

Fig. 1: Concept and realization of configuration-designable nanoscale three-dimensional printing.

Original fig. 2:

Fig. 2: Mechanism and fabrication details of two-dimensional nanotransfer printing on the elastomer substrate.

➔

Revised fig. 2:

[Response to the Reviewers #3’ Comments]

General Comments to the Author:

In this manuscript, the authors describe a new fabrication method to produce mechanically guided assembly of buckled nanostructures. Even though the pre-strained-substrate-induced buckling is not a brand-new concept, the reported fabrication method is novel and versatile and can enable deterministic buckling of nanoscale 2D patterns. The work is comprehensive

and presented in a clear and logical way with detailed experimental procedures and extensive discussion. There are a few major strength and novelty of this work. (1) Using an innovative pattern transfer technique, this work extended the feature size limit of previously reported mechanically guided buckling systems. The technique appears to be relatively simply and reliable up to 10s of nanometers. (2) It is very important that the authors did thorough mechanical analysis and simulation to deterministically predict and reversely design the buckling direction and mode. This really broadens the scope of the work. (3) The materials chemistry is quite versatile, which would be crucial for potential applications. The authors proposed two approaches to transfer nano-materials with different adhesion strength. (4) The authors also demonstrated a proof-of-concept application of the buckled nano-patterns as strain-insensitive gas sensors.

Overall, the reviewer believes this manuscript is of the high quality of other papers on Nature Communications, if the following issues can be addressed. First and foremost, the use of the term “4D printing” is inaccurate and confusing to a general audience. The reviewer understands that there are a number of papers on 3D printing processes with some post-printing structural transformation claiming the concept of “4D printing”. Even though this terminology is not a consensus in the field, the reviewer thinks it is acceptable if and only if (1) 3D printing is used to create 3D structures with a large degree of geometric freedom and (2) such 3D structure can be transformed after fabrication with an additional temporal degree of freedom (3D + 1D). However, in the context of this work, the authors used conventional nanofabrication techniques to create 2D patterns, transferred them to a pre-strained substrate, and then released the substrate to create 3D structures—this process is referred to in the manuscript as “4D printing”. The constant reference to 2D, 3D, and 4D appears to be quite confusing for readers. There are a lot of kirigami-inspired works that deform specific cut planar patterns in 3D (Choi, W. J. et al. Nat. Mater. 18, 820–826 (2019) for example). Should we consider them “4D printing” as well? The authors also refer to the pattern transferring process as printing, which is not accurate in a more modern sense of the word. To the reviewer’s understanding, printing (ink-jet printing in 2D or 3D printing) involves processes that can create/define patterns and structures on the fly. For example, an ink-jet printer can print any documents the user sends to it instead of just transferring/copying an existing document. Therefore, the reviewer would recommend reconsidering the use of “4D printing” and rephrasing the pattern transferring process “imprinting” or other more appropriate terms instead of “printing” as used in the manuscript.

Reply: We greatly appreciate your insightful suggestions. First, regarding the term ‘4D’, we agree that the word is not accurate and can be confusing to the general audience. As the reviewer mentioned, we used the term of ‘4D printing’ because some recent papers reporting technologies based on mechanically assembly claimed so [R1, R2]. In these studies, authors may define 4D printing in a broad sense as ‘conventional printing + transformation over time (additional shape-morphing by external stimuli)’. However, after careful reconsideration, we concluded that the definition from the reviewer ‘(1) 3D printing is used to create 3D structures with a large degree of geometric freedom and (2) such 3D structure can be transformed after fabrication with an additional temporal degree of freedom (3D + 1D).’ is more accurate and universal, and technologies based on mechanically guided assembly are just a kind of the advanced 3D printings. Thus, we have changed ‘4D printing’ to ‘3D printing’ in the entire manuscript including the title, as per the reviewer’s suggestion, and some representative changes are as follows (we attached a few examples here, all changes can be clearly seen in the ‘Manuscript_with_highlight’ file).

Second, regarding the term of ‘nanotransfer printing’, we also agree that in a rigorous way,

nanotransfer process cannot be defined as printing because the pattern cannot be directly designed in real-time as the reviewer discussed. However, in the field of nanofabrication, the term of ‘nanotransfer printing’ is customarily used by researchers in most of recent and high-quality research papers (even with the abbreviation ‘nTP’) [R3, R4, R5]. Furthermore, of the term ‘nanoimprint lithography’ which is totally different process with the nanotransfer printing is already in use [R6]. Therefore, we decide to use ‘nanotransfer printing’ as a technical and customary term to avoid misunderstanding and confusion from the general audience.

Title

Original text: “Nanoscale Four-Dimensional Printing Based on Mechanically Guided Assembly”

Revised text: “Nanoscale **Three**-Dimensional Printing Based on Mechanically Guided Assembly”

Introduction

Original text: “Given the difficulty of fabricating complex 3D structures using conventional two-dimensional (2D) planar processes such as inkjet/screen printing and lithography/ion-milling-based techniques, 3D and the recently developed four-dimensional (4D) printing methods have drawn much attention. In particular, 4D printing methods based on mechanically guided assembly (i.e., compressive buckling-based printing) allow the fabrication of complex 3D structures in thin and curvilinear forms and offer the advantages of wide material scope (e.g., metals, semiconductors, and polymers), high designability, precise controllability, scalability, and structural reversibility under strain, thus holding great promise for next-generation printing”

Revised text: “Given the difficulty of fabricating complex 3D structures using conventional two-dimensional (2D) planar processes such as inkjet/screen printing and lithography/ion-milling-based techniques, **the recently developed and more advanced 3D printing methods** have drawn much attention. In particular, **3D printing methods based on mechanically guided assembly** (i.e., compressive buckling-based printing) allow the fabrication of complex 3D structures in thin and curvilinear forms and offer the advantages of wide material scope (e.g., metals, semiconductors, and polymers), high designability, precise controllability, scalability, and structural reversibility under strain, thus holding great promise for next-generation printing.”

References

R1. Zhu, H. *et al.* Mechanically-Guided 4D Printing of Magnetoresponse Soft Materials across Different Length Scale. *Advanced Intelligent Systems* **4**, 3, 2100137 (2022).

- R2. Taylor, J. M. *et al.* Biomimetic and Biologically Compliant Soft Architectures via 3D and 4D Assembly Methods: A Perspective. *Advanced Materials* **34**, 16, 2108391 (2022).
- R3. Park, T. W. *et al.* Thermally assisted nanotransfer printing with sub–20-nm resolution and 8-inch wafer scalability. *Science advances* **6**, 31, eabb6462 (2020).
- R4. Zhao, Z.-J. *et al.* Direct Chemisorption-Assisted Nanotransfer Printing with Wafer-Scale Uniformity and Controllability. *ACS nano* **16**, 1, 378-385 (2022).
- R5. Ko, J. *et al.* Nanotransfer printing on textile substrate with water-soluble polymer nanotemplate. *ACS nano* **14**, 2, 2191-2201 (2020).
- R6. Guo, L. J. Nanoimprint lithography: methods and material requirements. *Advanced materials* **19**, 4, 495-513 (2007).

Some specific comments:

[**Comment 1**] It would be good to briefly explain what is Mode 1/2/3 buckling to general audience.

Reply: We appreciate your valuable comment and agree that brief explanation for buckling mode 1/2/3 should be added for improving clarity. Thus, we have made modifications as per the reviewer's suggestion in Results and discussion as follows.

Results and discussion

Original text: “In addition, as shown in Fig. 3d-iii, the configuration of the single beam (measured using CLSM (Supplementary Fig. 15)) was well-matched with the results of the FEM simulation. Subsequently, we investigated the buckling mode (Fig. 3e), which is determined and controlled by choosing appropriate printing boundary conditions, and developed a design and fabrication method for controlling beam constraints. At a width (w_{pillar}) above 10 μm , the bound site enables beam fixation, as the thick pillar limits both beam displacement and beam rotation. On the contrary, $w_{\text{pillar}} < 3 \mu\text{m}$ corresponds to a pinned condition that limits only beam displacement and allows beam rotation, as fixation by a thin (compared with beam length) pillar is similar to point fixing (Fig. 3e-ii).”

Revised text: “In addition, as shown in Fig. 3d-iii, the configuration of the single beam (measured using CLSM (Supplementary Fig. 15)) was well-matched with the results of the FEM simulation. Subsequently, we investigated the buckling modes (Fig. 3e), which describe the shape of the beam when the buckling occurs (to intuitively explain it, mode 1 has no inflection point, mode 2 has one inflection point, and mode 3 has two inflection points). Generally, buckling with mode 1 occurs most frequently because the required critical force increases as the mode number increases (i.e., when a gradually increasing compressive force is applied to a beam, buckling with mode 1 occurs first before generating mode 2 or 3 under higher compressive force). However, buckling with higher mode can be

controlled by adding mechanical constraints hindering buckling with lower mode. Thus, we developed a design and fabrication method for controlling the beam constraints by choosing appropriate printing boundary conditions, enabling the design of buckling mode. At a width (w_{pillar}) above $10\ \mu\text{m}$, the bound site enables beam fixation, as the thick pillar limits both beam displacement and beam rotation. On the contrary, $w_{\text{pillar}} < 3\ \mu\text{m}$ corresponds to a pinned condition that limits only beam displacement and allows beam rotation, as fixation by a thin (compared with beam length) pillar is similar to point fixing (Fig. 3e-ii)."

[Comment 2] In Fig. 2e, there are some white dots on the edges of the ribbons. What is the cause? Are they related to O₂ plasmas treatment? Would H₂ plasma solve some of the oxidation issues without using protective coating?

Reply: We appreciate your valuable comment and helpful idea. The white dots on the edges of the nanoribbons were caused by a side-deposition during e-beam evaporation process. As shown in Figs. 3e and R4, target material with a few nanometers was deposited on the side wall of nanopillars in the mold, and after transfer process, it appears as white dots in the top-view SEM image. Therefore, the dots are observed only on the edges of the nanoribbons, and this issue can be solved by improving vacuum state of the e-beam evaporation equipment. However, regardless of the white dots, oxidation issues exist because of the O₂ plasma treatment with long time, hindering perfect printing of easily oxidized materials such as Ag and Fe without protective layer as shown in Supplementary Fig. 5. Although short-term O₂ plasma treatment less than 10 s is required to activate the hydroxyl groups for chemical bonding at the surface, replacement of long-term O₂ plasma treatment to H₂ plasma treatment as suggested by reviewer is expected to be a great solution to solve the oxidation issue because H₂ plasma treatment can also selectively etch carbon atoms of substrate without oxidation, and thus, we have added them with appropriate reference to the results and discussion section as follows.

Fig. 3e. SEM images of nanoline patterns with linewidths of 50 nm and 2 μm nanotransfer-printed on a flat elastomer substrate

Fig. R4. Schematic illustration for explaining occurring of white dots in SEM images.

Results and discussion

Original text: “Notably, our method featured a broad scope of transferable materials (e.g., metals and ceramics) and was applicable to the transfer of multilayer structures (Supplementary Fig. 4). Therefore, materials easily oxidized during treatment with O₂ plasma (e.g., Ag and Fe) could be transferred without oxidation or corrosion via encapsulation with noble metals before printing (Supplementary Fig. 5).”

Revised text: “Notably, our method featured a broad scope of transferable materials (e.g., metals and ceramics) and was applicable to the transfer of multilayer structures (Supplementary Fig. 4). Therefore, materials easily oxidized during treatment with O₂ plasma (e.g., Ag and Fe) could be transferred without oxidation or corrosion via encapsulation with noble metals before printing (Supplementary Fig. 5). **It is also expected that these materials can be transferred with H₂ plasma treatment without oxidation²⁰.**”

Reference

20. Xu, J. *et al.* Different Etching Mechanisms of Diamond by Oxygen and Hydrogen Plasma: A Reactive Molecular Dynamics Study. *Journal of Physical Chemistry C* **125**, 16711–16718 (2021).

[Comment 3] A few sentences explaining the general principles of the electromechanical gas sensor would be helpful to readers.

Reply: We appreciate your valuable suggestion, and agree that adding an explanation about the working principles of the gas sensors will be valuable for clearer understanding. Thus, we have made modifications as per the reviewer’s suggestion to Supplementary Notes 4 as follows.

Results and discussion

Original text: “Similar to the H₂ sensor, it maintained the response to 10 ppm NO₂ with a variance of 12.5% when an external strain of up to 30% was sequentially applied. It reveals that various gas sensors with the different detection gases can be fabricated using the developed nanoscale 4D printing”

Revised text: “Similar to the H₂ sensor, it maintained the response to 10 ppm NO₂ with a variance of 12.5% when an external strain of up to 30% was sequentially applied. It reveals

that various gas sensors with the different detection gases can be fabricated using the developed nanoscale 3D printing. The detailed explanation for working mechanisms of each gas sensor is described in Supplementary Notes 4”

Supplementary Notes 4

Added text: “Supplementary Notes 4: Working mechanisms of gas sensors. First, for Pd-based H₂ gas sensor, Pd is converted to palladium hydride (PdH_x) by absorbing H₂ gas under normal temperature and pressure conditions. Therefore, when Pd is exposed to H₂ gas, the electrical resistance change and mechanical deformation (volume expansion) occur because PdH_x has higher electrical resistance and larger volume than pure Pd. In this study, the electrical resistance was used as a monitoring parameter to detect H₂ gas. Second, for In₂O₃-based NO₂ gas sensor, when n-type semiconductor material (i.e., In₂O₃ in this study) are thermal- or photo-activated (i.e., photo-activation in this study) and exposed to oxidizing gas (i.e., NO₂ in this study), oxygen is adsorbed on the nanostructure surface by capturing free electrons. Thus, the depletion layer of semiconductor material enlarges and electrical resistance increases. In this study, NO₂ gas concentration was monitored by measuring the electrical resistance of In₂O₃.”

[Comment 4] More discussion on why the field needs to scale down the mechanically guided buckling to nm scale would make the motivation section stronger.

Reply: We appreciate your helpful comment and agree that the discussion about motivation is not described in detail in the main manuscript to avoid repeated sentences. Although designed 3D nanostructures are in high demand in various applications such as nanophotonic devices, sensors, and energy-related devices, nanoscale fabrication is still challenging for most existing printing methods. We supposed that mechanically guided assembly-based fabrication technology has great potential in fabrication of 3D nanostructures because of its superior controllability and simplicity. We have already described them in Supplementary Notes 1. The related texts are attached below for your reference.

Supplementary Notes 1

Related text: “Summary of recently developed 3D printing based on mechanically guided assembly. As discussed in the main text, 3D printing methods based on mechanically guided assembly have been actively studied to improve design diversity with various materials¹⁻⁹ and to develop new applications¹⁰⁻¹⁶ or inverse design techniques¹⁷. Although 3D nanostructures are in high demand¹⁸⁻²¹, nanoscale fabrication is still challenging for most existing printing methods including mechanically guided assembly-based 4D printing. Hence, we herein considered the printable size range as one of the core problems to be solved.”

“It is meaningful to reduce the width of the bound site as it is directly related to the design diversity and number of printable devices per unit area, and the suspended site shows width-dependent chemomechanical properties, as exemplified by nanowires with ultra-high surface area to volume ratios^{22,23}.”

Introduction

Related text: “Thus, the realization of configuration-designable nanoscale 3D printing based on mechanically guided assembly remains challenging, and the demand for 3D nanodevices for applications such as gas sensors, electrodes, thermoacoustic speakers, and optical devices remains unmet¹⁸.”

REVIEWERS' COMMENTS

Reviewer #1 (Remarks to the Author):

I appreciate the authors' efforts in responding to my comments.

Reviewer #3 (Remarks to the Author):

The revised manuscript and the response letter address my comments well. This is an interesting and technically solid article worthy of publication in Nature Communications. The only remaining concern I have is the use of "3D printing" to describe the mechanically guidance assembly process. I'm glad the authors and the other reviewer agreed on the inaccuracy of using "4D printing" to describe the process. "3D printing" is also confusing because people generally use it to describe processes that produce a large variety of geometries. I would highly recommend use "3D fabrication" in the title and other places since this is a more accurate, to my opinion. Thanks for your consideration.

REVIEWER REPORT:

[Response to Reviewers #3' Comments]

Comments to the Author:

The revised manuscript and the response letter address my comments well. This is an interesting and technically solid article worthy of publication in Nature Communications. The only remaining concern I have is the use of "3D printing" to describe the mechanically guidance assembly process. I'm glad the authors and the other reviewer agreed on the inaccuracy of using "4D printing" to describe the process. "3D printing" is also confusing because people generally use it to describe processes that produce a large variety of geometries. I would highly recommend use "3D fabrication" in the title and other places since this is a more accurate, to my opinion. Thanks for your consideration.

Reply: We sincerely appreciate your careful review and valuable comments on our manuscript. We agree that the term '3D printing' should be changed to '3D fabrication' for clear understanding of the readers. Thus, we have changed '3D printing' to '3D fabrication' in the entire manuscript including the title as follows (all changes can be clearly seen in the 'Manuscript_with_highlight' file). We hope that the revised manuscript is suitable for publication in *Nature Communications*.

Title

Original text: "Nanoscale Three-Dimensional Printing Based on Mechanically Guided Assembly"

Revised text: "Nanoscale Three-Dimensional **Fabrication** Based on Mechanically Guided Assembly"